# The Primal-Dual method for Learning Augmented Algorithms

**Etienne Bamas**[*]
EPFL, Lausanne, Switzerland
etienne.bamas@epfl.ch

**Andreas Maggiori**[*]
EPFL, Lausanne, Switzerland
andreas.maggiori@epfl.ch

**Ola Svensson**[*]
EPFL, Lausanne, Switzerland
ola.svensson@epfl.ch

## Abstract

The extension of classical online algorithms when provided with predictions is a new and active research area. In this paper, we extend the primal-dual method for online algorithms in order to incorporate predictions that advise the online algorithm about the next action to take. We use this framework to obtain novel algorithms for a variety of online covering problems. We compare our algorithms to the cost of the true and predicted offline optimal solutions and show that these algorithms outperform any online algorithm when the prediction is accurate while maintaining good guarantees when the prediction is misleading.

## 1 Introduction

In the classical field of online algorithms the input is presented in an online fashion and the algorithm is required to make irrevocable decisions without knowing the future. The performance is often measured in terms of worst-case guarantees with respect to an optimal offline solution. In this paper, we will consider minimization problems and formally, we will say that an online algorithm $\mathcal{ALG}$ is *c-competitive* if on any input $\mathcal{I}$, the cost $c_{\mathcal{ALG}}(\mathcal{I})$ of the solution output by algorithm $\mathcal{ALG}$ on input $\mathcal{I}$ satisfies $c_{\mathcal{ALG}}(\mathcal{I}) \leqslant c \cdot \mathrm{OPT}(\mathcal{I})$, where $\mathrm{OPT}(\mathcal{I})$ denotes the cost of the offline optimum. Due to the uncertainty about the future, online algorithms tend to be overly cautious which sometimes causes their performance in real-world situations to be far from what a machine learning (ML) algorithm would have achieved. Indeed in many practical applications future events follow patterns which are easily predictable using ML methods. In [19] Lykouris and Vassilvitskii formalized a general framework for incorporating (ML) predictions into online algorithms and designed an extension of the marking algorithm to solve the online caching problem when provided with predictions. This work was quickly followed by many other papers studying different learning augmented online problems such as scheduling ([17]), caching ([2, 26]), ski rental ([10, 16, 25, 28]), clustering ([7]) and other problems ([12, 23]). The main challenge is to incorporate the prediction without knowing how the prediction was computed and in particular without making any assumption on the quality of the prediction. This setting is natural as in real-world situations, predictions are provided by ML algorithms that rarely come with worst-case guarantees on their accuracy. Thus, the difficulty in designing a learning augmented algorithm is to find a good balance: on the one hand, following blindly the prediction might lead to a very bad solution if the prediction is misleading. On the other hand if the algorithm does not trust the prediction at all, it will simply never benefit from an excellent prediction. The aforementioned results solve this issue by designing smart algorithms

---

[*]Equal Contribution.

which exploit the problem structure to achieve a good trade-off between these two cases. In this paper we take a different perspective. Instead of focusing on a specific problem trying to integrate predictions, we show how to extend a very powerful algorithmic method, the Primal-Dual method, into the design of online learning augmented algorithms. We underline that despite the generality of our extension technique, it produces online learning augmented algorithms in a fairly simple and straightforward manner.

**The Primal-Dual method.** The Primal-Dual (PD) method is a very powerful algorithmic technique to design online algorithms. It was first introduced by Alon et al. [1] to design an online algorithm for the classical online set cover problem and later extended to many other problems such as weighted caching ([3]), revenue maximization in ad-auctions, TCP acknowledgement and ski rental [5]. We mention the survey of Buchbinder and Naor [4] for more references about this technique. In a few words, the technique consists in formulating the online problem as a linear program $P$ complemented by its dual $D$. Subsequently, the algorithm builds online a feasible fractional solution to both the primal $P$ and dual $D$. Every time an update of the primal and dual variables is made, the cost of the primal increases by some amount $\Delta P$ while the cost of the dual increases by some amount $\Delta D$. The competitive ratio of the fractional solution is then obtained by upper bounding the ratio $\frac{\Delta P}{\Delta D}$ and using weak duality. The integral solution is then obtained by an online rounding scheme of the fractional solution.

**Preliminary notions for Learning Augmented (LA) algorithms.** LA algorithms receive as input a prediction $\mathcal{A}$, an instance $\mathcal{I}$ which is revealed online, a robustness parameter $\lambda$, and output a solution of cost $c_{\mathcal{ALG}}(\mathcal{A}, \mathcal{I}, \lambda)$. Intuitively, $\lambda$ indicates our confidence in the prediction with smaller values reflecting high confidence. We denote by $S(\mathcal{A}, \mathcal{I})$ the cost of the output solution on input $\mathcal{I}$ if the algorithm follows blindly the prediction $\mathcal{A}$. We avoid defining explicitly prediction $\mathcal{A}$ to easily fit different prediction cases. For instance if the prediction $\mathcal{A}$ is a predicted solution (without necessarily revealing the predicted instance) then following blindly the solution would simply mean to output the predicted solution $\mathcal{A}$. For each result presented in this paper, it will be clear what is the prediction $\mathcal{A}$ and the cost $S(\mathcal{A}, \mathcal{I})$. Given this, we restate some useful definitions introduced in [19, 25] in our context. For any $0 < \lambda \leqslant 1$, we will say that an LA algorithm is $C(\lambda)$-*consistent* and $R(\lambda)$-*robust* if the cost of the output solution satisfies:

$$c_{\mathcal{ALG}}(\mathcal{A}, \mathcal{I}, \lambda) \leqslant \min\{C(\lambda) \cdot S(\mathcal{A}, \mathcal{I}), R(\lambda) \cdot \text{OPT}(\mathcal{I})\} \tag{1}$$

If $\mathcal{A}$ is accurate ($S(\mathcal{A}, \mathcal{I}) \approx \text{OPT}(\mathcal{I})$) and at the same time we trust the prediction, we would like our performance to be close to the optimal offline. Thus, ideally $C(\lambda)$ should approach 1 as $\lambda$ approaches 0. On the same spirit, a value of $\lambda$ close to 1 denotes no trust to the prediction, and in that case, our algorithm should not be much worse than the best pure online algorithm. Therefore, $R(1)$ should be close to the competitive ratio of the best pure online algorithm. We also mention that in some other papers such as [25], a *smoothness* criterion on the consistency bound is required. In these papers the setting is slightly different. Indeed, prediction $\mathcal{A}$ is a predicted instance $\mathcal{I}^{pred}$ and the error is defined to describe how far $\mathcal{I}^{pred}$ is from the real instance $\mathcal{I}$. With this in mind, an algorithm is said to be *smooth* if the performance degrades smoothly as the error increases. We emphasize that, in the applications considered in this paper, this smoothness property is implicitly included in the value of $S(\mathcal{A}, \mathcal{I})$ which degrades smoothly with the quality of the prediction.

**Our contributions.** We show how to extend the Primal-Dual method (when predictions are provided) for solving problems that can be formulated as covering problems. The algorithms designed using this technique receive as input a robustness parameter $\lambda$ and incorporate a prediction. If the prediction is accurate our algorithms can be arbitrarily close to the optimal offline (beating known lower bounds of the classical online algorithms) while being robust to failures of the predictor. We first apply our Primal-Dual Learning Augmented (PDLA) technique to the online version of the weighted set cover problem, which constitutes the most canonical example of a covering Linear Program (LP). For that problem we show how we can easily modify the Primal-Dual algorithm to incorporate predictions. Even though in this case, prediction may not seem very natural, this result reveals that we can use PDLA to design learning augmented algorithms for the large class of problems that can be formulated as a covering LP. We then continue by addressing problems in which the prediction model is much more natural. Using the PDLA technique, we first design an algorithm which recovers the results of Purohit et al. [25] for the ski rental problem, and we also prove that the consistency-robustness trade-off of that algorithm is optimal. We additionally design a

learning augmented algorithm for a generalization of the ski rental, namely the Bahncard problem. Finally, we turn our attention to a problem which arises in network congestion control, the TCP acknowledgement problem. We design an LA algorithm for that problem and conduct experiments which confirm our claims. We note that the analysis of the algorithms designed using PDLA is (arguably) simple and boils down to (1) proving robustness with (essentially) the same proof as in the original Primal-Dual technique and (2) proving consistency using a simple charging argument that, without making use of the dual, relates the cost incurred by our algorithms to the prediction. In addition to that, using PDLA, the design of online LA algorithms is almost automatic. We emphasize that the preexisting online rounding schemes to obtain an integral solution from a fractional solution still apply to our learning augmented algorithms. Hence in all the paper we focus only on building a fractional solution and provide appropriate references for the rounding scheme.

## 2   General PDLA method

In this section we apply PDLA to solve the online weighted set cover problem when provided with predictions. Set cover is arguably the most canonical example of a covering problem and the framework that we develop readily applies to other covering problems. In particular, we use the framework to give tight or nearly-tight LA algorithms for ski rental, Bahncard, and dynamic TCP acknowledgement, which are all problems that can be formulated as covering LPs.

**The weighted set cover problem.**   In this problem, we are given a universe $\mathcal{U} = \{e_1, e_2, \ldots, e_n\}$ of $n$ elements and a family $\mathcal{F}$ of $m$ sets over this universe, each set $S \in \mathcal{F}$ has a weight $w_S$ and each element $e$ is covered by any set in $\mathcal{F}(e) = \{S \in \mathcal{F} \mid e \in S\}$. Let $d = \max_{e \in \mathcal{U}} |\mathcal{F}(e)|$ denote the maximum number of sets that cover one element. Our goal is to select sets so as to cover all elements while minimizing the total weight. In its online version, elements are given one by one and it is unknown to the algorithm which elements will arrive and in which order. When a new element arrives, it is required to cover it by adding a new set if necessary. Removing a set from the current solution to decrease its cost is not

| Primal |
|---|
| minimize $\sum_{S \in \mathcal{F}} w_S x_S$ |
| subject to: $\sum_{S \in \mathcal{F}(e)} x_S \geq 1 \quad \forall e \in \mathcal{U}$ |
| $x_S \geq 0 \quad \forall S \in \mathcal{F}$ |
| **Dual** |
| maximize $\sum_{e \in \mathcal{U}} y_e$ |
| subject to: $\sum_{e \in S} y_e \leq w_S \quad \forall S \in \mathcal{F}$ |
| $y_e \geq 0 \quad \forall e \in \mathcal{U}$ |

Figure 1: Primal Dual formulation of weighted set cover

allowed. Alon et al. in [1] first studied the online version designing an almost optimal $O(\log n \log d)$-competitive algorithm. We note that the $O(\log n)$ factor comes from the integrality gap of the linear program formulation of the problem (Figure 1) while the $O(\log d)$ is due to the online nature of the problem. Since Alon et al. [1] designed an online rounding scheme at a multiplicative cost of $O(\log n)$, we will focus on building an increasing fractional solution to the set cover problem (i.e. $x_S$ can only increase over time for all $S$).

**PDLA for weighted set cover.**   Algorithm 2 takes as input a predicted covering $\mathcal{A} \subset \mathcal{F}$ and a robustness parameter $\lambda \in [0, 1]$. While an instance $\mathcal{I}$ is revealed in an online fashion, an increasing fractional solution $\{x_S\}_{S \in \mathcal{F}} \in [0, 1]^{\mathcal{F}}$ is built. Note that $\mathcal{F}(e) \cap \mathcal{A}$ are the sets which cover $e$ in the prediction. To simplify the description, we assume that $|\mathcal{F}(e) \cap \mathcal{A}| \geqslant 1, \forall e$, i.e. the prediction forms a feasible solution. The algorithm without this assumption can be found in the supplemental material.

**Algorithm Intuition.**   We first turn our attention to the original online algorithm of Alon et. al. [1] described in Algorithm 1. To get an intuition assume that $w_S = 1, \forall S$ and consider the very first arrival of an element $e$. After the first execution of the while loop, $e$ is covered and $x_S = \frac{1}{|\mathcal{F}(e)|}, \forall S \in \mathcal{F}(e)$. In other words, the online algorithm creates a uniform distribution over the sets in $\mathcal{F}(e)$, reflecting in such a way his unawareness about the future. On the contrary Algorithm 2 uses the prediction to adjust the increase rate of primal variables, augmenting more aggressively primal variables of sets which are predicted to be in the optimal offline solution. Indeed, after the first execution of the while loop, sets which belong to $\mathcal{A}$ get a value of $\frac{\lambda}{|\mathcal{F}(e)|} + \frac{1-\lambda}{|\mathcal{F}(e) \cap \mathcal{A}|}$ while sets which are not chosen by the prediction get $\frac{\lambda}{|\mathcal{F}(e)|}$.

| **Algorithm 1** PRIMAL DUAL METHOD FOR ONLINE WEIGHTED SET COVER [1]. |
|---|

**Initialize:** $x_S \leftarrow 0, y_e \leftarrow 0 \ \forall S, e$
**for all** element $e$ that just arrived **do**
   **while** $\sum_{S \in \mathcal{F}(e)} x_S < 1$ **do**
      /* Primal Update
      **for all** $S \in \mathcal{F}(e)$ **do**
         $x_S \leftarrow x_S \left(1 + \frac{1}{w_S}\right) + \frac{1}{w_S |\mathcal{F}(e)|}$
      **end for**
      /* Dual Update
      $y_e \leftarrow y_e + 1$
   **end while**
**end for**

$\Rightarrow$

| **Algorithm 2** PDLA FOR ONLINE WEIGHTED SET COVER. |
|---|

**Input:** $\lambda, \mathcal{A}$
**Initialize:** $x_S \leftarrow 0, y_e \leftarrow 0 \ \forall S, e$
**for all** element $e$ that just arrived **do**
   **while** $\sum_{S \in \mathcal{F}(e)} x_S < 1$ **do**
      /* Primal Update
      **for all** $S \in \mathcal{F}(e)$ and $S \in \mathcal{A}$ **do**
         $x_S \leftarrow x_S \left(1 + \frac{1}{w_S}\right) + \frac{\lambda}{w_S |\mathcal{F}(e)|} + \frac{1-\lambda}{w_S |\mathcal{F}(e) \cap \mathcal{A}|}$
      **end for**
      **for all** $S \in \mathcal{F}(e)$ and $S \notin \mathcal{A}$ **do**
         $x_S \leftarrow x_S \left(1 + \frac{1}{w_S}\right) + \frac{\lambda}{w_S |\mathcal{F}(e)|}$
      **end for**
      /* Dual Update
      $y_e \leftarrow y_e + 1$
   **end while**
**end for**

We continue by exposing our main conceptual contribution. To that end let $S(\mathcal{A}, \mathcal{I})$ denote the cost of the covering solution described by prediction $\mathcal{A}$ on instance $\mathcal{I}$.

**Theorem 1.** *Assuming $\mathcal{A}$ is a feasible solution, the cost of the fractional solution output by Algorithm 2 satisfies*

$$c_{\mathcal{PDLA}}(\mathcal{A}, \mathcal{I}, \lambda) \leqslant \min \left\{ O\left(\frac{1}{1-\lambda}\right) \cdot S(\mathcal{A}, \mathcal{I}), O\left(\log\left(\frac{d}{\lambda}\right)\right) \cdot \mathrm{OPT}(\mathcal{I}) \right\}$$

*Proof sketch.* The proof is split in two parts. The first part is to bound the cost of the algorithm by the term $O\left(\frac{1}{1-\lambda}\right) \cdot S(\mathcal{A}, \mathcal{I})$. As mentioned in the introduction we use a charging argument to do so. After each execution of the while loop we can decompose the primal increase into two parts. $\Delta P_c$ which denotes the increase due to sets in $\mathcal{F}(e) \cap \mathcal{A}$ and $\Delta P_u$ which denotes the increase due to sets in $\mathcal{F}(e) \setminus \mathcal{A}$, thus for the overall primal increase $\Delta P$ we have $\Delta P = \Delta P_c + \Delta P_u$. We continue by upper bounding $\Delta P_u$ as a function of $\lambda$ and $\Delta P_c$, that is $\Delta P_u \leqslant O(\frac{1+\lambda}{1-\lambda})\Delta P_c$, and deducing that $\Delta P \leqslant O(\frac{1}{1-\lambda})\Delta P_c$. Now the consistency proof terminates by noting that since $\Delta P_c$ is generated by sets in the prediction, we can charge this increase to $S(\mathcal{A}, \mathcal{I})$. The robustness bound, which is independent of the prediction, is retrieved by mimicking the proof of the original online algorithm of Alon et al. [1]. See the supplemental material for more details. $\square$

## 3 The Ski rental problem

As another application of PDLA we design a learning augmented algorithm for one of the simplest and well studied online problems, the ski rental problem. In this problem, every new day, one has to decide whether to rent skis for this day, which costs 1 dollar or to buy skis for the rest of the vacation at a cost of $B$ dollars. In its offline version the total number of vacation days, $N$, is known in advance and the problem becomes trivial. From the primal-dual formulation of the problem (Figure 2) it is clear that if $B < N$, the optimal strategy is to buy the skis at day one while if $B \geqslant N$ the optimal strategy is to always rent. In the online setting the difficulty relies in the fact that we do not know $N$ in advance. A deterministic 2-competitive online algorithm has been known for a long time [13] and a randomized $\frac{e}{e-1} \approx 1.58$-competitive algorithm was also designed later [14]. Both competitive ratios are known to

| **Primal** |
|---|
| minimize $B \cdot x + \sum_{j \in [N]} f_j$ |
| subject to: $x + f_j \geq 1 \quad \forall j \in [N]$ |
| $x, f_j \geq 0 \quad \forall j \in [N]$ |
| **Dual** |
| maximize $\sum_{j \in [N]} y_j$ |
| subject to: $\sum_{j \in [N]} y_j \leq B$ |
| $1 \geq y_j \geq 0 \quad \forall j \in [N]$ |

Figure 2: Primal dual formulation of the ski rental problem.

be optimal for deterministic and randomized algorithms respectively. This problem was already studied in various learning augmented settings [10, 16, 25, 28]. Our approach recovers, using the primal-dual method, the results of [25]. As in [25] our prediction $\mathcal{A}$ will be the total number of vacation days $N^{pred}$.

**PDLA for ski rental.** To simplify the description, we denote an instance of the problem as $\mathcal{I} = (N, B)$ and define the function $e(z) = (1 + 1/B)^{z \cdot B}$. Note that if $B \to \infty$, then $e(z)$ approaches $e^z$ hence the choice of notation. In an integral solution, the variable $x$ is 1 to indicate that the skis are bought and 0 otherwise. In the same spirit $f_j$ indicates whether we rent on day $j$ or not. Buchbinder et al. [5] showed how to easily turn a fractional monotone solution (i.e. it is not permitted to decrease a variable) to an online randomized algorithm of expected cost equal to the cost of the fractional solution. Hence we focus only on building online a fractional solution. Algorithm 3 is due to [5] and uses the Primal-Dual method to solve the problem. Each new day $j$ a new constraint $x + f_j \geqslant 1$ is revealed. To satisfy this constraint, the algorithm updates the primal and dual variables while trying to maintain (1) the ratio $\Delta P / \Delta D$ as small as possible and (2) the primal and dual solutions feasible. As in the online weighted set cover problem, the key idea for extending Algorithm 3 to the learning augmented Algorithm 4 is to use the prediction $N^{pred}$ in order to adjust the rate at which each variable is increased. Thus, when $N^{pred} > B$ we increase the buying variable more aggressively than the pure online algorithm. Here, the cost of following blindly the prediction $N^{pred}$ is $S(N^{pred}, \mathcal{I}) = B \cdot \mathbb{1}\{N^{pred} > B\} + N \cdot \mathbb{1}\{N^{pred} \leqslant B\}$.

---

**Algorithm 3** PRIMAL DUAL FOR SKI-RENTAL [5].

---

**Initialize:** $x \leftarrow 0$, $f_j \leftarrow 0$, $\forall j$
$c \leftarrow e(1)$, $c' \leftarrow 1$
**for** each new day $j$ s.t. $x + f_j < 1$ **do**
  /* Primal Update
  $f_j \leftarrow 1 - x$
  $x \leftarrow (1 + \frac{1}{B})x + \frac{1}{(c-1) \cdot B}$
  /* Dual Update
  $y_j \leftarrow c'$
**end for**

---

$\Longrightarrow$

---

**Algorithm 4** PDLA FOR SKI-RENTAL.

---

**Input:** $\lambda$, $N^{pred}$
**Initialize:** $x \leftarrow 0$, $f_j \leftarrow 0$, $\forall j$
**if** $N^{pred} \geqslant B$ **then**
  /* Prediction suggests buying
  $c \leftarrow e(\lambda)$, $c' \leftarrow 1$
**else**
  /* Prediction suggests renting
  $c \leftarrow e(1/\lambda)$, $c' \leftarrow \lambda$
**end if**
**for** each new day $j$ s.t. $x + f_j < 1$ **do**
  /* Primal Update
  $f_j \leftarrow 1 - x$
  $x \leftarrow (1 + \frac{1}{B})x + \frac{1}{(c-1) \cdot B}$
  /* Dual Update
  $y_j \leftarrow c'$
**end for**

---

In the following we assume that either $\lambda B$ or $B/\lambda$ is an integer (depending on whether $c$ equals $e(\lambda)$ or $e(1/\lambda)$ respectively in Algorithm 4). Our results do not change qualitatively by rounding up to the closest integer. See the supplemental material for details.

**Theorem 2** (PDLA for ski rental). *For any $\lambda \in (0, 1]$, the cost of PDLA for ski rental is bounded as follows*

$$c_{\mathcal{PDLA}}(N^{pred}, \mathcal{I}, \lambda) \leqslant \min \left\{ \frac{\lambda}{1 - e(-\lambda)} \cdot S(N^{pred}, \mathcal{I}), \frac{1}{1 - e(-\lambda)} \cdot \mathrm{OPT}(\mathcal{I}) \right\}$$

*Proof sketch.* The robustness bound is proved essentially using the same proof as for the original analysis of Algorithm 3 in [5]. For the consistency bound we first note that after an update the primal increase is $1 + \frac{1}{c-1}$, now depending on the value of $c$ we distinguish between two cases. If $N^{pred} \geqslant B$ then Algorithm 4 is always aggressive in buying. In this case it is easy to show that at most $\lambda B$ updates are made before we get $x \geqslant 1$. Once $x \geqslant 1$, no more updates are needed. Since each aggressive update costs at most $1 + \frac{1}{e(\lambda)-1} = \frac{e(\lambda)}{e(\lambda)-1} = \frac{1}{1-e(-\lambda)}$ we get that the total cost paid by Algorithm 4 is at most $\frac{\lambda B}{1-e(-\lambda)} = S(N^{pred}, \mathcal{I}) \cdot \frac{\lambda}{1-e(-\lambda)}$. Similarly, in the second case $N^{pred} < B$ and the algorithm increases the buying variable less aggressively. In this case each update costs at

most $1 + \frac{1}{e(1/\lambda)-1} = \frac{1}{1-e(-1/\lambda)}$ and at most $N$ of these updates are made therefore Algorithm 4 pays at most $\frac{N}{1-e(-1/\lambda)} = S(N^{pred}, \mathcal{I}) \cdot \frac{1}{1-e(-1/\lambda)}$. To conclude the consistency proof, note that $\frac{1}{1-e(-1/\lambda)} \leqslant \frac{\lambda}{1-e(-\lambda)}$ (see the supplemental material). □

In addition to recovering the positive results of [25], we additionally show in the supplemental material that this consistency-robustness trade-off is optimal.

**Lemma 3.** *Any $\frac{\lambda}{1-e^{-\lambda}}$-consistent learning augmented algorithm for ski rental has robustness $R(\lambda) \geqslant \frac{1}{1-e^{-\lambda}}$*

To emphasize how PDLA permits us to tackle more general problems, we apply the same ideas to a generalization of the ski-rental problem, namely, the Bahncard problem [9]. This problem models a situation where a tourist travels every day multiple trips. Before any new trip, the tourist has two choices, either to buy a ticket for that particular trip at a cost of 1 or buy a discount card, at a cost of $B$, that allows to buy tickets at a cheaper price of $\beta < 1$. The discount card remains valid during $T$ days. Note that ski-rental is modeled by taking $\beta = 0$ and $T \to \infty$. In the learning augmented version of the problem we are given a prediction $\mathcal{A}$ which consists in a collection of times where we are advised to acquire the discount card. We state the main result on this problem and defer the proof to the supplemental material.

**Theorem 4** (PDLA for the Bahncard problem). *For any $\lambda \in (0, 1]$, any $\beta \in [0, 1]$ and $\frac{B}{1-\beta} \to \infty$, we have the following guarantees on any instance $\mathcal{I}$ and prediction $\mathcal{A}$*

$$cost_{\mathcal{PDLA}}(\mathcal{A}, \mathcal{I}, \lambda) \leqslant \min\left\{ \frac{\lambda}{1-\beta+\lambda\beta} \cdot \frac{e^\lambda - \beta}{e^\lambda - 1} \cdot S(\mathcal{A}, I), \frac{e^\lambda - \beta}{e^\lambda - 1} \cdot \text{OPT}(\mathcal{I}) \right\}$$

## 4 Dynamic TCP acknowledgement

In this section, we continue by applying PDLA to a classic network congestion problem of the Transmission Control Protocol (TCP). During a TCP interaction, a server receives a stream of packets and replies back to the sender acknowledging that each packet arrived correctly. Instead of sending an acknowledgement for each packet separately, the server can choose to delay its response and acknowledge multiple packets simultaneously via a single TCP response. Of course, in this scenario there is an additional cost incurred due to the delayed packets, which is the total latency incurred by those packets. Thus, on one hand sending too many acknowledgments (acks) overloads the network, on the other hand sending one ack for all the packets slows down the TCP interaction. Hence a good trade-off has

| **Primal** |
| --- |
| minimize $\sum_{t \in T} x_t + \sum_{j \in M} \sum_{t \mid t \geq t(j)} \frac{1}{d} f_{jt}$ |
| subject to: $f_{jt} + \sum_{k=t(j)}^{t} x_k \geq 1 \quad \forall j, t \geq t(j)$ |
| $f_{jt} \geq 0 \quad \forall j, t \geq t(j)$ |
| $x_t \geq 0 \quad \forall t \in T$ |
| **Dual** |
| maximize $\sum_{j \in M} \sum_{t \mid t \geq t(j)} y_{jt}$ |
| subject to: $\sum_{j \mid t \geq t(j)} \sum_{t' \geq t} y_{jt} \leq 1 \quad \forall t \in T$ |
| $0 \leq y_{jt} \leq \frac{1}{d} \quad \forall j, t \geq t(j)$ |

Figure 3: Primal Dual formulation of the TCP acknowledgement problem

to be achieved and the objective function which we aim to minimize will be the sum of the total number of acknowledgements plus the total latency. The problem was first modeled by Dooly et al. [8], where they showed how to solve the offline problem optimally in quadratic time along with a deterministic 2-competitive online algorithm. Karlin et al. [15] provided the first $\frac{e}{e-1}$-competitive randomized algorithm which was later shown to be optimal by Seiden in [27]. The problem was later solved using the primal-dual method by Buchbinder et al. [5] who also obtained an $\frac{e}{e-1}$-competitive algorithm. Figure 3 presents the primal-dual formulation of the problem. In this formulation each packet $j$ arrives at time $t(j)$ and is acknowledged by the first ack sent after $t(j)$. Here, variable $x_t$ corresponds to sending an ack at time $t$ and $f_{jt}$ is set to one (in the integral solution) if packet $j$ was not acknowledged by time $t$. The time granularity is controlled by the parameter $d$ and each additional time unit of latency comes at a cost of $1/d$. As in the ski rental problem, there is no integrality gap and a fractional monotone solution can be converted to a randomized algorithm in a lossless manner (see [5] for more details).

## 4.1 The PDLA algorithm and its theoretical analysis

Our prediction consists in a collection of times $\mathcal{A}$ in which the prediction suggests sending an ack. Let $\alpha(t)$ be the next time $t' \geqslant t$ when prediction sends an ack. With this definition each packet $j$, if the prediction is followed blindly, is acknowledged at time $\alpha(t(j))$ incurring a latency cost of $(\alpha(t(j)) - t(j)) \cdot \frac{1}{d}$. In the same spirit as for the ski rental problem we adapt the pure online Algorithm 5 into the learning augmented Algorithm 6. Algorithm 6 adjusts the rate at which we increase the primal and dual variables according to the prediction $\mathcal{A}$. Thus if a packet $j$ at time t is "uncovered" ($\sum_{k=t(j)}^{t} x_k + f_{jt} < 1$) by our fractional solution and "covered" by $\mathcal{A}$ ($\alpha(t(j)) \leqslant t$) we increase $x_t$ at a faster rate. To simplify the description of Algorithm 6 we define $e(z) = (1 + \frac{1}{d})^{z \cdot d}$. To get to the continuous time case, we will take the limit $d \to \infty$ so the reader should think intuitively as $e(z) \approx e^z$.

**Algorithm 5** PRIMAL DUAL METHOD FOR TCP ACKNOWLEDGEMENT [5].

> **Initialize:** $x \leftarrow 0, y \leftarrow 0$
> **for all** times $t$ **do**
>> **for all** packages $j$ such that $\sum_{k=t(j)}^{t} x_k < 1$ **do**
>>> $c \leftarrow e(1), c' \leftarrow 1/d$
>>> /* Primal Update
>>> $f_{jt} \leftarrow 1 - \sum_{k=t(j)}^{t} x_k$
>>> $x_t \leftarrow x_t + \frac{1}{d} \cdot \left( \sum_{k=t(j)}^{t} x_k + \frac{1}{c-1} \right)$
>>> /* Dual Update
>>> $y_{jt} \leftarrow c'$
>> **end for**
> **end for**

$\Longrightarrow$

**Algorithm 6** PDLA FOR TCP ACKNOWLEDGEMENT

> **Input:** $\lambda, \mathcal{A}$
> **Initialize:** $x \leftarrow 0, y \leftarrow 0$
> **for all** times $t$ **do**
>> **for all** packages $j$ such that $\sum_{k=t(j)}^{t} x_k < 1$ **do**
>>> **if** $t \geqslant \alpha(t(j))$ **then**
>>>> /* Prediction already acknowledged packet $j$
>>>> $c \leftarrow e(\lambda), c' \leftarrow 1/d$
>>> **else**
>>>> /* Prediction did not acknowledge packet $j$ yet
>>>> $c \leftarrow e(1/\lambda), c' \leftarrow \lambda/d$
>>> **end if**
>>> /* Primal Update
>>> $f_{jt} \leftarrow 1 - \sum_{k=t(j)}^{t} x_k$
>>> $x_t \leftarrow x_t + \frac{1}{d} \cdot \left( \sum_{k=t(j)}^{t} x_k + \frac{1}{c-1} \right)$
>>> /* Dual Update
>>> $y_{jt} \leftarrow c'$
>> **end for**
> **end for**

We continue by presenting Algorithm's 6 guarantees together with a proof sketch. As before $\mathcal{I}$ denotes the TCP ack problem instance which is revealed in an online fashion. The full proof is deferred to the supplemental material.

**Theorem 5** (PDLA for TCP-ack). *For any prediction $\mathcal{A}$, any instance $\mathcal{I}$ of the TCP ack problem, any parameter $\lambda \in (0,1]$, and $d \to \infty$: Algorithm 6 outputs a fractional solution of cost at most*

$$c_{\mathcal{PDLA}}(\mathcal{A}, \mathcal{I}, \lambda) \leqslant \min \left\{ \frac{\lambda}{1 - e^{-\lambda}} \cdot S(\mathcal{A}, \mathcal{I}), \frac{1}{1 - e^{-\lambda}} \cdot \text{OPT}(\mathcal{I}) \right\}$$

*Proof sketch.* The two bounds are proven separately. For the robustness bound, while our analysis is slightly more technical, we use the same idea as the original analysis in [5]. That is, upper bounding the ratio $\Delta P / \Delta D$ in every iteration and using weak duality. The consistency proof uses a simple charging scheme that can be seen as a generalization of our consistency proof for the ski rental problem. We essentially have two cases, big ($c = e(\lambda)$) and small ($c = e(1/\lambda)$) updates. In the case of a small update, a simple calculation reveals that the increase in cost of the solution is at most $\Delta P = \frac{1}{d} \left( 1 - \sum_{k=t(j)}^{t} x_k \right) + \frac{1}{d} \left( \sum_{k=t(j)}^{t} x_k + \frac{1}{e(1/\lambda)-1} \right) = \frac{1}{d} \left( 1 + \frac{1}{e(1/\lambda)-1} \right) = \frac{1}{d} \cdot \left( \frac{1}{1 - e(-1/\lambda)} \right)$. Notice then whenever Algorithm 6 does a small update at time $t$ due to request $j$, prediction $\mathcal{A}$ pays a latency cost of $1/d$ since it has not yet acknowledged request $j$. Hence the primal increase of cost which is at most $\frac{1}{d} \cdot \frac{1}{1 - e(-1/\lambda)}$ can be charged to the latency cost $1/d$ paid by $\mathcal{A}$ with a multiplicative factor $\frac{1}{1 - e(-1/\lambda)} \leqslant \frac{\lambda}{1 - e(-\lambda)}$ (see the supplemental material). The case of big updates is slightly different. Consider a time $t_0$ at which $\mathcal{A}$ sends an acknowledgement and consider the big updates performed by Algorithm 6 for packets $j$ arrived before that time ($t(j) \leqslant t_0$). We claim that at most $\lceil \lambda d \rceil$ such big updates can be made. Indeed, big updates are more aggressive (i.e. $x_t$ increases

faster), and a "covering" due to $\sum_{k=t_0}^{t} x_k \geqslant 1$ is reached after only $\lceil \lambda d \rceil$ updates (after this point, the packets arrived before time $t_0$ will never force Algorithm 6 to make an update). Thus Algorithm's 6 cost due to these big updates is at most $\lceil \lambda d \rceil \cdot$ (cost of a big update) $= \lceil \lambda d \rceil \cdot (\frac{1}{d} \cdot \frac{\lambda}{1-e(-\lambda)})$ which can be charged to the cost of 1 incurred by $\mathcal{A}$ for sending an ack at time $t_0$. $\qquad\square$

## 4.2 Experiments

We present experimental results that confirm the theoretical analysis of Algorithm 6 for the TCP acknowledgement problem. The code is publicly available at `https://github.com/etienne4/PDLA`. We experiment on various types of distribution for packet arrival inputs. Historically, the distribution of TCP packets was often assumed to follow some Poisson distribution ([20, 30]). However, it was later shown than this assumption was not always representative of the reality. In particular real-world distributions often exhibit a heavy tail (i.e. there is still a significant probability of seeing a huge amount of packets arriving at some time). To better integrate this in models, heavy tailed distributions such as the Pareto distribution are often suggested (see for instance [11, 22]). This motivates our choice of distributions for random packet arrival instances. We will experiment on Poisson distribution, Pareto distribution and a custom distribution that we introduce and seems to generate the most challenging instances for our algorithms.

**Input distributions.**    In all our instances, we set the subdivision parameter $d$ to 100 which means that every second is split into 100 time units. Then we define an array of length 1000 where the $i$-th entry defines how many requests arrive at the $i$-th time step. Each entry in the array is drawn independently from the others from a distribution $\mathcal{D}$. In the case of a Poisson distribution, we set $\mathcal{D} = \mathcal{P}(1)$ (the Poisson distribution of mean 1). For the Pareto distribution, we choose $\mathcal{D}$ to be the Lomax distribution (which is a special case of Pareto distribution) with shape parameter set to 2 ([29]). Finally, we define the *iterated* Poisson distribution as follows. Fix an integer $n > 0$ and $\mu > 0$. Draw $X_1 \sim \mathcal{P}(\mu)$. Then for $i$ from 2 to $n$ draw $X_i \sim \mathcal{P}(X_{i-1})$. The final value returned is $X_n$. This distribution, while still having an expectation of $\mu$, appears to generate more spikes than the classical Poisson distribution. The interest of this distribution in our case is that it generates more challenging instances than the other two (i.e. the competitive ratios of online algorithms are closer to the worst-case bounds). In our experiments, we choose $\mu = 1$ and $n = 10$. Plots of typical instances under these laws can be seen in the supplemental material. Note that for all these distributions, the expected value for each entry is 1.

**Noisy prediction.**    The prediction $\mathcal{A}$ is produced as follows. We perturb the real instances with noise, then compute an optimal solution on this perturbed instance and use this as a prediction. More precisely, we introduce a *replacement* rate $p \in [0, 1]$. Then we go through the instance generated according to some distribution $\mathcal{D}$ and for each each entry at index $1 \leqslant i \leqslant 1000$, with probability $p$ we set this entry to 0 (i.e. we delete this entry) and with probability $p$ we add to this entry a random variable $Y \sim \mathcal{D}$. Both operations, adding and deleting, are performed independently of each other. We then test our algorithm with 4 different values of robustness parameter $\lambda \in \{1, 0.8, 0.6, 0.4\}$.

**Results.**    The plots in Figure 4 present the average competitive ratios of Algorithm 6 over 10 experiments for each distribution and each value of $\lambda$. As expected, with a perfect prediction, setting a lower $\lambda$ will yield a much better solution while setting $\lambda = 1$ simply means that we run the pure online algorithm of Buchbinder et al. [5] (that achieves the best possible competitive ratio for the pure online problem). On the most challenging instances generated by the iterated Poisson distribution (Figure 4c), even with a replacement rate of 1 where the prediction is simply an instance totally uncorrelated to the real instance, our algorithm maintains good guarantees for small values of $\lambda$. We note that in all the experiments the competitive ratios achieved by Algorithm 6 are better than the robustness guarantees of Theorem 5, which are $\{1.58, 1.68, 2.21, 3.03\}$ for $\lambda \in \{1, 0.8, 0.6, 0.4\}$ respectively. In addition to that, all the competitive ratios degrade smoothly as the error increases which confirms our earlier discussion about smoothness.

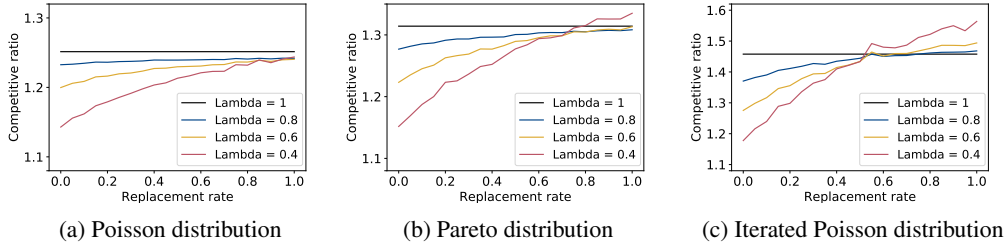

|                        |                         |                                   |
| :--------------------: | :---------------------: | :-------------------------------: |
| (a) Poisson distribution | (b) Pareto distribution | (c) Iterated Poisson distribution |

Figure 4: Competitive ratios under various distributions and replacement rates from 0 to 1

## 5 Future Directions

In this paper we present the PDLA technique, a learning augmented version of the classic Primal-Dual technique, and apply it to design algorithms for some classic online problems when a prediction is provided. Since the Primal-Dual technique is used to solve many more covering problems, like for instance weighted caching or load balancing [4], an interesting research direction would be to apply PDLA to tackle those problems and (hopefully) get tight consistency-robustness trade-off guarantees (as the one achieved by Algorithm 4 and proved in Lemma 3). In addition to that, we suspect that this work might provide insights not only for covering but also for some packing problems which are solved using the Primal-Dual technique in the classic online model (e.g. revenue maximization in ad-auctions [5]). Finally, another interesting direction would be to incorporate predictions into the Primal-Dual technique when used to solve covering problems where the objective function is non linear (e.g. convex).

## Broader Impact

The field of learning augmented algorithms lies in the intersection of machine learning and online algorithms, trying to combine the best of the two worlds. Learning augmented algorithms are particularly suited for critical applications where maintaining worst-case guarantees is mandatory but at the same time predictions about the future are possible. Thus, our work represents a stepping stone towards (easily) integrating ML predictions in such applications, increasing this way the possible benefits of ML to society. PDLA offers a recipe on how to incorporate predictions to tackle classical covering online problems, that is to first solve the online problem using the Primal-Dual technique and then use the prediction to change the rate at which primal and dual variables increase or decrease. We believe that since the idea behind this technique is simple and does not require too much domain-specific knowledge, it might be applicable to different problems and can also be implemented in practice.

## Acknowledgments and Disclosure of Funding

This research is supported by the Swiss National Science Foundation project 200021-184656 "Randomness in Problem Instances and Randomized Algorithms". Andreas Maggiori was supported by the Swiss National Science Fund (SNSF) grant nº 200020_182517/1 "Spatial Coupling of Graphical Models in Communications, Signal Processing, Computer Science and Statistical Physics".

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
