[Supplementary Material]

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

# A    Missing proofs for Set Cover

We first present the slightly modified algorithm where we do not need the prediction to form a feasible solution. As mentioned in the main paper, when an element $e$ is uncovered by the prediction, i.e. $|\mathcal{F}(e) \cap \mathcal{A}| = 0$, we just run the purely online algorithm ($\lambda = 1$).

---

**Algorithm 7** PDLA FOR ONLINE WEIGHTED SET COVER.

---

   **Input:** $\lambda$, $\mathcal{A}$
   **Initialize:** $x_S \leftarrow 0$, $y_e \leftarrow 0 \,\forall S, e$
   **for all** element $e$ that just arrived **do**
      **while** $\sum_{S \in \mathcal{F}(e)} x_S < 1$ **do**
         **for all** $S \in \mathcal{F}(e)$ **do**
            **if** $|\mathcal{F}(e) \cap \mathcal{A}| \geqslant 1$ **then**
               /* Primal Update (more aggressive if $\mathbb{1}\{S \in \mathcal{A}\} = 1$ )
               $x_S \leftarrow x_S \left(1 + \frac{1}{w_S}\right) + \frac{\lambda}{w_S \cdot |\mathcal{F}(e)|} + \frac{(1-\lambda) \cdot \mathbb{1}\{S \in \mathcal{A}\}}{w_S \cdot |\mathcal{F}(e) \cap \mathcal{A}|}$
            **else**
               /* e is not covered by the prediction
               $x_S \leftarrow x_S \cdot \left(1 + \frac{1}{w_S}\right) + \frac{1}{w_S \cdot |\mathcal{F}(e)|}$
            **end if**
         **end for**
         /* Dual Update
         $y_e \leftarrow y_e + 1$
      **end while**
   **end for**

---

We start by proving that the dual constraints are only violated by a multiplicative factor of $O\left(\log\left(\frac{d}{\lambda}\right)\right)$. Thus, scaling down the dual solution of Algorithm 7 by $O\left(\log\left(\frac{d}{\lambda}\right)\right)$ creates a feasible dual solution which will permit us to use weak duality.

**Lemma 6.** *Let $y$ be the dual solution built by Algorithm 7. Then $\frac{y}{\Theta(\log(d/\lambda))}$ is a feasible solution to the dual problem.*

*Proof.* The proof essentially follows the same path as in [4]. The only constraints that can be violated are of the form $\sum_{e \in S} y_e \leqslant w_S$ for some $S \in \mathcal{F}$. Consider one such constraint. At every update of the primal variable $x_S$ the sum $\sum_{e \in S} y_e$ increases by 1, since the dual variable corresponding to the newly arrived element increases by 1 . We prove by induction on the number of such updates that at any point in time $x_S \geqslant \frac{\lambda}{d}\left(\left(1 + \frac{1}{w_S}\right)^{\sum_{e \in S} y_e} - 1\right)$. Indeed, when no update concerning $S$ is done we have that $x_S = 0$ and $\sum_{e \in S} y_e = 0$. Suppose this is true after $k$ updates of the variable $x_S$, i.e. $\sum_{e \in S} y_e = k$. Now, assume that a newly arrived element $e^* \in S$ provokes a primal update from $x_S^{old}$ to $x_S^{new}$ and increases its dual value by one, i.e. $y_{e^*}^{new} = y_{e^*}^{old} + 1$. Then we always have:

$$
\begin{aligned}
x_S^{new} &\geqslant x_S^{old} \cdot \left(1 + \frac{1}{w_S}\right) + \min\left\{\frac{1}{|\mathcal{F}(e)| \cdot w_S}, \frac{\lambda}{|\mathcal{F}(e)| \cdot w_S} + \frac{(1-\lambda) \cdot \mathbb{1}\{S \in \mathcal{A}\}}{|\mathcal{F}(e) \cap \mathcal{A}| \cdot w_S}\right\} \geqslant \\
&\geqslant x_S^{old} \cdot \left(1 + \frac{1}{w_S}\right) + \frac{\lambda}{d \cdot w_S}
\end{aligned}
$$

Thus, by the induction hypothesis

$$
\begin{aligned}
x_S^{new} &\geqslant \frac{\lambda}{d}\left(\left(1 + \frac{1}{w_S}\right)^{\sum_{e \in S \setminus \{e^*\}} y_e + y_{e^*}^{old}} - 1\right) \cdot \left(1 + \frac{1}{w_S}\right) + \frac{\lambda}{d \cdot w_S} \\
&= \frac{\lambda}{d}\left(\left(1 + \frac{1}{w_S}\right)^{\sum_{e \in S \setminus \{e^*\}} y_e + y_{e^*}^{new}} - 1\right) = \frac{\lambda}{d}\left(\left(1 + \frac{1}{w_S}\right)^{\sum_{e \in S} y_e} - 1\right)
\end{aligned}
$$

Moreover, since $w_S \geqslant 1$, we have that $(1 + 1/w_s)^{w_s} \geqslant 2$, thus:

$$x_S \geqslant \frac{\lambda}{d}\left(\left(1 + \frac{1}{w_S}\right)^{w_S \cdot \frac{\sum_{e \in S} y_e}{w_S}} - 1\right) \geqslant \frac{\lambda}{d}\left(2^{\frac{\sum_{e \in S} y_e}{w_S}} - 1\right)$$

We continue by upper bounding the value of $x_S$. Note that once $x_S \geqslant 1$, no more primal updates can happen, therefore whenever an update is made we have $x_S < 1$ just before the update. Thus:

$$x_S^{new} \leqslant x_S^{old} \cdot \left(1 + \frac{1}{w_S}\right) + \max\left\{\frac{\lambda}{w_S \cdot |\mathcal{F}(e)|} + \frac{(1-\lambda) \cdot \mathbb{1}\{S \in \mathcal{A}\}}{w_S \cdot |\mathcal{F}(e) \cap \mathcal{A}|}, \frac{1}{w_S \cdot |\mathcal{F}(e)|}\right\}$$

$$\leqslant x_S^{old} \cdot 2 + 1 \leqslant 3$$

Combining the lower and upper bound on $x_S$ we get that:

$$\sum_{e \in S} y_e \leqslant \log\left(\frac{3d}{\lambda} + 1\right) \cdot w_S = O(\log(d/\lambda)) \cdot w_S$$

which concludes the proof. $\qquad\square$

**Lemma 7** (Robustness). *The competitive ratio is always bounded by $O\left(\log\left(\frac{d}{\lambda}\right)\right)$*

*Proof.* We denote as before by $x_S^{old}$ and $x_S^{new}$ the primal variables before and after the update respectively. Each time the while loop is executed we have that $\sum_{S \in \mathcal{F}(e)} x_S^{old} < 1$ and the increase in the dual is $\Delta D = 1$. Denote by $\delta x_S = x_S^{new} - x_S^{old}$ the increase of a variable for a specific set $S$. If an element is covered by the prediction then it holds that:

$$\Delta P = \sum_{S \in \mathcal{F}(e)} w_S \cdot \delta x_S = \sum_{S \in \mathcal{F}(e) \cap \mathcal{A}} w_S \cdot \delta x_S + \sum_{S \in \mathcal{F}(e) \backslash \mathcal{F}(e) \cap \mathcal{A}} w_S \cdot \delta x_S =$$

$$= \sum_{S \in \mathcal{F}(e)} \left(x_S^{old} + \frac{\lambda}{|\mathcal{F}(e)|}\right) + \sum_{S \in \mathcal{F}(e) \cap \mathcal{A}} \frac{(1-\lambda)}{|\mathcal{F}(e) \cap \mathcal{A}|} = \sum_{S \in \mathcal{F}(e)} x_S^{old} + \lambda + 1 - \lambda \leqslant 2$$

By repeating the same calculation we get that if an element is uncovered by the prediction then:

$$\Delta P = \sum_{S \in \mathcal{F}(e)} w_S \cdot \delta x_S = \sum_{S \in \mathcal{F}(e)} \left(x_S^{old} + \frac{1}{|\mathcal{F}(e)|}\right) = \sum_{S \in \mathcal{F}(e)} x_S^{old} + 1 \leqslant 2$$

Overall we have that:

1. At any iteration $\frac{\Delta P}{\Delta D} \leqslant 2$.

2. The final primal solution is feasible.

3. By Lemma 6, denoting $y$ the final dual solution, $\frac{y}{\Theta(\log(d/\lambda))}$ is feasible.

Thus, by weak duality we get that the competitive ratio of Algorithm 7 is upper bounded by $2 \cdot O(\log(d/\lambda)) = O(\log(d/\lambda)))$. $\qquad\square$

In the following we do not assume that our prediction $\mathcal{A}$ forms a feasible solution. Therefore we will denote by

1. $S(\mathcal{A}, \mathcal{I})$ the cost of the (possibly partial) covering if prediction $\mathcal{A}$ is followed blindly.

2. $C_{nc}$ the cost of optimally covering elements which are not covered by the prediction.

3. $c_{\mathcal{PDLA}}(\mathcal{A}, \mathcal{I}, \lambda)$ the cost of the covering solution calculated by Algorithm 7.

**Lemma 8** (Consistency). $c_{\mathcal{PDLA}}(\mathcal{A}, \mathcal{I}, \lambda) \leqslant O\left(\frac{1}{1-\lambda}\right) \cdot S(\mathcal{A}, \mathcal{I}) + O(\log(d)) \cdot C_{nc}$

*Proof.* We split the analysis in two parts. First, we look at the case when an element which is uncovered by the prediction arrives. In this case Algorithm 7 emulates the pure online algorithm ($\lambda = 1$). More precisely, by the same calculations as before, we can show that $y_{nc}$ the solution of the dual problem restricted to the uncovered elements satisfy the property that $\frac{y_{nc}}{O(\log d)}$ is feasible. Therefore for those elements by Lemma 7 the cost of Algorithm 7 is upper bounded by $O(\log d) \cdot C_{nc}$. We turn our attention to the more interesting case where the prediction covers an element. In this case, after the execution of the while loop we decompose the primal increase into two parts. $\Delta P_c$ which denotes the increase due to sets $S$ chosen by $\mathcal{A}$ ($\mathbb{1}\{S \in \mathcal{A}\} = 1$) and $\Delta P_u$ which denotes the increase due to sets $S$ not chosen by the prediction ($\mathbb{1}\{S \in \mathcal{A}\} = 0$), thus we have $\Delta P = \Delta P_c + \Delta P_u$. Let $c = \{S \in \mathcal{F}(e) : \mathbb{1}\{S \in \mathcal{A}\} = 1\}$ and $u = \{S \in \mathcal{F}(e) : \mathbb{1}\{S \in \mathcal{A}\} = 0\}$. We then have:

$$\Delta P_c = \sum_{S \in c} x_S + \frac{\lambda \cdot |c|}{|c| + |u|} + 1 - \lambda \geqslant \frac{\lambda}{d} + 1 - \lambda$$

$$\Delta P_u = \sum_{S \in u} x_S + \frac{\lambda \cdot |u|}{|c| + |u|} \leqslant 1 + \lambda$$

since , $\frac{|c|}{|c|+|u|} \geqslant \frac{1}{d}$ and $\frac{|u|}{|c|+|u|} \leqslant 1$. Combining the two bounds we get that $\Delta P_u \leqslant \frac{1+\lambda}{\frac{\lambda}{d}+1-\lambda} \cdot \Delta P_c$ and consequently:

$$\Delta P \leqslant \left(1 + \frac{1+\lambda}{\frac{\lambda}{d} + 1 - \lambda}\right) \Delta P_c = O\left(\frac{1}{1-\lambda}\right) \Delta P_c$$

Since the cost increase $\Delta P_c$ is caused by sets which are selected by the prediction, we can charge this cost to the corresponding increase of $S(\mathcal{A}, \mathcal{I})$ loosing only a multiplicative $O(1)$ factor. By combining the two cases we conclude the proof. $\square$

## B    Missing proofs for ski rental and the Bahncard problem

We detail here the missing proofs from section 3. We first prove our results regarding the ski rental problem and then focus on the Bahncard problem.

### B.1    The ski rental problem

We provide here a full proof of Theorem 2. In our setting, the prediction $\mathcal{A}$ is the predicted number of skiing days $N^{pred}$ and $S(\mathcal{A}, \mathcal{I}) = S(N^{pred}, \mathcal{I}) = B \cdot \mathbb{1}\{N^{pred} > B\} + N \cdot \mathbb{1}\{N^{pred} \leqslant B\}$ is the cost of following blindly the prediction. We first prove an easy lemma about the feasibility of the dual solution.

**Lemma 9.** *Let $y$ be the dual solution built by Algorithm 4. Then $y$ is a feasible solution (assuming $\frac{B}{\lambda}$ is integral if the prediction suggests to rent).*

*Proof.* To see this, note that the only constraint that might by violated is the constraint $\sum_{j \in [N]} y_j \leqslant B$. Denote by $S$ the value of the sum $\sum_{j \in [N]} y_j$. Note that once $x \geqslant 1$, the value of $S$ will never change anymore. The value of $S$ increases by 1 for every big update and by $\lambda$ for every small update. In the case $N^{pred} > B$, the algorithm always does big updates (the prediction suggest to buy). We claim that at most $\lceil \lambda B \rceil$ big updates can be made before $x \geqslant 1$. We denote $x(k)$ the value of $x$ after $k$ updates. We then prove by induction that $x(k) \geqslant \frac{e(k/B)-1}{e(\lambda)-1}$ (recall that $e(z) = (1 + 1/B)^{z \cdot B} \approx e^z$).

Clearly, if $k = 0$, we have $x(0) \geqslant 0$. Now assume this is the case for $k$ updates we then have

$$
\begin{aligned}
x(k + 1) &= \left(1 + \frac{1}{B}\right) \cdot x(k) + \frac{1}{(e(\lambda) - 1) \cdot B} \\
&\geqslant \left(1 + \frac{1}{B}\right) \cdot \frac{e(k/B) - 1}{e(\lambda) - 1} + \frac{1}{(e(\lambda) - 1) \cdot B} \\
&= \frac{(1 + 1/B) \cdot (e(k/B) - 1) + 1/B}{e(\lambda) - 1} \\
&= \frac{e((k + 1)/B) - 1}{e(\lambda) - 1}
\end{aligned}
$$

which ends the induction. Hence at most $\lceil \lambda B \rceil \leqslant B$ big updates can be made before $x \geqslant 1$. This implies that $S \leqslant B$ at the end of the algorithm. In the case where $N^{pred} \leqslant B$, we prove in exactly the same way that at most $\left\lceil \frac{B}{\lambda} \right\rceil$ updates are performed before $x \geqslant 1$. Hence we have that $S \leqslant \lambda \cdot \left\lceil \frac{B}{\lambda} \right\rceil$. By assumption, we have that $B/\lambda$ is an integer hence $S \leqslant 1$ and $y$ is again feasible. □

We can finish the main proof.

*Proof of Theorem 2.* We prove first the robustness bound. By the Lemma 9, we know that the dual solution is feasible. Hence what remains to prove is to upper bound the ratio $\frac{\Delta P}{\Delta D}$ and use weak duality. In the case of a big update we have

$$
\frac{\Delta P}{\Delta D} = \Delta P = 1 + \frac{1}{e(\lambda) - 1} = \frac{1}{1 - e(-\lambda)}
$$

In the case of a small update we have

$$
\frac{\Delta P}{\Delta D} = \frac{\Delta P}{\lambda} = \frac{1}{\lambda} \cdot \frac{1}{1 - e(-1/\lambda)} \leqslant \frac{1}{1 - e(-\lambda)}
$$

where the last inequality comes from Lemma 19 inequality (2). By weak duality, we have the robustness bound.

To prove consistency, we have two cases. If $N^{pred} \leqslant B$, then Algorithm 4 does at most $N$ updates, each of cost at most $\frac{1}{1 - e(-1/\lambda)}$ while the prediction $\mathcal{A}$ pays a cost of $N$. Noting again that, by Lemma 19, $\frac{1}{1 - e(-1/\lambda)} \leqslant \frac{\lambda}{1 - e(-\lambda)}$ ends the proof of consistency in this case. The other case is different. As in the proof of Lemma 9, we still have that $x(k) \geqslant \frac{e(k/B) - 1}{e(\lambda) - 1}$ hence at most $\lceil \lambda B \rceil \leqslant B$ updates are done by Algorithm, each of cost at most $\frac{1}{1 - e(-\lambda)}$ hence a total cost of at most

$$
\frac{\lceil \lambda B \rceil}{1 - e(-\lambda)}
$$

Since we assume in this case that $\lambda B$ is integral and that the prediction $\mathcal{A}$ pays a cost of $B$, the competitive ratio is indeed $\frac{\lambda}{1 - e(-\lambda)}$ □

## B.2 The Bahncard problem

**History of the problem.** The Banhcard problem, which was initially introduced in [9], models a situation where a tourist travels every day multiple trips. Before any new trip, the tourist has two choices, either to buy a ticket for that particular trip at a cost of $1$ or buy a discount card, at a cost of $B$, and use this discount card to get a ticket for a price of $\beta < 1$. The discount card is then valid for rest of that day and for the next $T - 1$ days. This generalizes the ski rental problem in several ways, first the discount expires after a fixed amount of time, second buying only offers a discount and not a free trip. Note that if $\beta = 0$ and $T \to \infty$ we recover the ski-rental problem. Karlin et al. [15] designed an optimal randomized online algorithm of competitive ratio $\frac{e}{e - 1 + \beta}$ when $B \to \infty$.

**PDLA for the Bahncard problem.** We design, using PDLA, a learning augmented algorithm for the Bahncard problem. The final goal is to prove Theorem 4. An interesting feature of our algorithm is that, as for the TCP ack problem, it does not need to be given the full prediction in advance. If Bahncards are bought by the prediction $\mathcal{A}$ at a set of times $\{t_1, t_2, \ldots, t_k\}$, the algorithm does not need to know before time $t_i$ that the Bahncard $i$ is bought. For instance we could think of the prediction of an employee of the station giving short-term advice to a traveller every time he shows up at the station.

We now give the primal dual formulation of the Bahncard problem along with its corresponding learning augmented algorithm. We mention that, to the best of our knowledge, no online algorithm using the primal-dual method was designed before. Hence the primal-dual formulation (Figure 5) of the problem is new. In an integral solution, we would have $x_t = 1$ if the solution buys a Bahncard at time $t$ and $x_t = 0$ otherwise. Then $f_j$ represents the fractional amount of trip $j$ done at time $t(j)$ that is bought at full price and $d_j$ the amount of the trip bought at discounted price. The first natural constraint is the one that says that each trip should be paid entirely either in discounted or full price, i.e. $d_j + f_j \geqslant 1$. We then have the constraint $\sum_{t=t(j)-T}^{t(j)} x_t \geqslant d_j$ that says that to be able to buy a ticket at discounted price, at least one Bahncard must have been bought in the last $T$ time steps.

Figure 5: Primal Dual formulation of the Bahncard problem.

| Primal | Dual |
|---|---|
| minimize $B \cdot \sum_{t \in T} x_t + \sum_{j \in M} \beta d_j + f_j$ | maximize $\sum_{j \in M} c_j$ |
| subject to: $\quad d_j + f_j \geqslant 1 \quad \forall j$ | subject to: $\quad c_j \leqslant 1 \quad \forall j$ |
| $\sum_{t=t(j)-T}^{t(j)} x_t \geqslant d_j \quad \forall j$ | $c_j - b_j \leqslant \beta \quad \forall j$ |
| $x_t \geqslant 0 \quad \forall t \in \mathcal{T}$ | $\sum_{j:t(j)-T \leqslant t \leqslant t(j)} b_j \leqslant B \quad \forall t \in \mathcal{T}$ |
| $d_j, f_j \geqslant 0 \quad \forall j$ | $c_j, b_j \geqslant 0 \quad \forall j$ |

Following the same idea as for the ski rental problem, we will guide the updates in the primal-dual algorithm with the advice provided. We define a function $e(z) = \left(1 + \frac{1-\beta}{B}\right)^{z \cdot (B/(1-\beta))}$. Again for $\frac{B}{1-\beta} \to \infty$, the reader should think intuitively of $e(z)$ as $e^z$. The parameter $z$ will then take values either $\lambda$ or $1/\lambda$ depending on if we want to do a big or small update in the primal. As for ski rental, when we do a small update, we will need to scale down the dual update by a factor of $\lambda$ to maintain feasibility of the dual solution.

The rule to decide if an update should be big or small is the following: if the prediction $\mathcal{A}$ bought a Bahncard less than $T$ time steps in the past (i.e. if the predicted solution has currently a valid Bahncard) the update should be big. Otherwise the update should be cautious. In algorithm 8, we denote by $l_{\mathcal{A}}(t)$ the latest time before time t at which the prediction $\mathcal{A}$ bought a Bahncard. We use the convention that $l_{\mathcal{A}}(t) = -\infty$ if no Bahncard was bought before time $t$. Of course in this problem it is possible that trips show up while the fractional solution already has a full Bahncard available (i.e. $\sum_{t=t(j)-T}^{t(j)} x_t \geqslant 1$). In this case there is no point in buying more fractions of a Bahncard and the algorithm will do what we call a *minimal* update.

---

**Algorithm 8** LA ONLINE PRIMAL-DUAL FOR THE BAHNCARD PROBLEM

---

**Input:** $\lambda$, $\mathcal{A}$
**Initialize:** $x, d, f \leftarrow 0$, $c, b \leftarrow 0$
**for all** trip $j$ **do**
    **if** $\sum_{t=t(j)-T}^{t(j)} x_t \geqslant 1$ **then**
        $d_j \leftarrow 1$
        $c_j \leftarrow \beta$
    **end if**
    **if** $\sum_{t=t(j)-T}^{t(j)} x_t < 1$ **then**
        **if** $t(j) \leqslant l_{\mathcal{A}}(t(j)) + T$ **then**
            $d_j \leftarrow \sum_{t=t(j)-T}^{t(j)} x_t$
            $f_j \leftarrow 1 - d_j$
            $x_{t(j)} \leftarrow x_{t(j)} + \frac{1-\beta}{B} \cdot \left( \sum_{t=t(j)-T}^{t(j)} x_t + \frac{1}{e(\lambda)-1} \right)$
            $b_j \leftarrow 1 - \beta$
            $c_j \leftarrow b_j + \beta$
        **end if**
        **if** $t(j) > l_{\mathcal{A}}(t(j)) + T$ **then**
            $d_j \leftarrow \sum_{t=t(j)-T}^{t(j)} x_t$
            $f_j \leftarrow 1 - d_j$
            $x_{t(j)} \leftarrow x_{t(j)} + \frac{1-\beta}{B} \cdot \left( \sum_{t=t(j)-T}^{t(j)} x_t + \frac{1}{e(1/\lambda)-1} \right)$
            $b_j \leftarrow \lambda(1 - \beta)$
            $c_j \leftarrow b_j + \beta$
        **end if**
    **end if**
**end for**

---

We first prove that the dual built by the algorithm is almost feasible.

**Lemma 10.** *Let $(c,b)$ be the dual solution built by Algorithm 8, then $\frac{(c,b)}{1+(1-\beta)/B}$ is feasible.*

*Proof.* Note that the constraints $c_j \leqslant 1$ and $c_j - b_j \leqslant \beta$ are clearly maintained by the algorithm. And scaling down both $c$ and $b$ by some factor bigger than 1 will not alter their feasibility. Hence we focus only on the constraints of the form $\sum_{j:t(j)-T\leqslant t\leqslant t(j)} b_j \leqslant B$ for a fixed time $t$. Note that during a minimal update, the value of $b_j$ is not changed hence only small or big updates can alter the value of the sum $\sum_{j:t(j)-T\leqslant t\leqslant t(j)} b_j$. Similarly as for proofs in ski rental, denote by $b$ the number of big updates that are counted in this sum and by $s$ the number of small updates in this sum.

We first notice that once we have that $\sum_{t'=t}^{t+T} x_{t'} \geqslant 1$, no updates that alter the constraint $\sum_{j:t(j)-T\leqslant t\leqslant t(j)} b_j$ can happen. To see this, note that upon arrival of a trip $j$ between time $t$ and $t+T$, we have $\sum_{t=t(j)-T}^{t(j)} x_t \geqslant \sum_{t'=t}^{t(j)} x_{t'} = \sum_{t'=t}^{t+T} x_{t'}$.

Denote by $S$ the value of the sum $\sum_{t'=t}^{t+T} x_{t'}$. Note that for a big update, we have that the value of the sum $S$ is increased to at least $S \cdot \left(1 + \frac{1-\beta}{B}\right) + \frac{1-\beta}{B} \cdot \frac{1}{e(\lambda)-1}$. Similarly for a small update the new value of the sum is at least $S \cdot \left(1 + \frac{1-\beta}{B}\right) + \frac{1-\beta}{B} \cdot \frac{1}{e(1/\lambda)-1}$. Hence we can apply directly Lemma 20 with $d = \frac{B}{1-\beta}$ to conclude that once $b + \lambda s \geqslant \frac{B}{1-\beta}$, we have that $S \geqslant 1$.

Since for a big update, the sum $\sum_{j:t(j)-T\leqslant t\leqslant t(j)} b_j$ increases by $1 - \beta$ and by $\lambda(1 - \beta)$ for a small update we can see that the first time the constraint $\sum_{j:t(j)-T\leqslant t\leqslant t(j)} b_j \leqslant B$ is violated, we have $S \geqslant 1$. Now since each update in the sum $\sum_{j:t(j)-T\leqslant t\leqslant t(j)} b_j$ is of value at most $1 - \beta$ we can conclude that at the end of the algorithm, we have $\sum_{j:t(j)-T\leqslant t\leqslant t(j)} b_j \leqslant B + 1 - \beta$ hence the conclusion. $\square$

We then prove robustness of Algorithm 8 by the following lemma.

**Lemma 11** (Robustness). *For any $\lambda \in (0, 1]$ and any $\beta \in [0, 1]$, PDLA for the Bahncard problem is $\frac{(e(\lambda) - \beta) \cdot (1 + (1 - \beta)/B)}{e(\lambda) - 1}$-robust.*

*Proof.* Algorithm 8 makes 3 possible types of updates. For a minimal update, we have $\Delta P = \Delta D = \beta$. For a small update we have

$$\Delta P = (1 - \beta) \cdot \left( \sum_{t=t(j)-T}^{t(j)} x_t + \frac{1}{e(1/\lambda) - 1} \right) + \beta \cdot \sum_{t=t(j)-T}^{t(j)} x_t + 1 - \sum_{t=t(j)-T}^{t(j)} x_t$$

$$= 1 + \frac{1 - \beta}{e(1/\lambda) - 1} = \frac{e(1/\lambda) - \beta}{e(1/\lambda) - 1}$$

and

$$\Delta D = \lambda(1 - \beta) + \beta = \beta(1 - \lambda) + \lambda$$

hence the ratio is

$$\frac{\Delta P}{\Delta D} = \frac{1}{\beta(1 - \lambda) + \lambda} \cdot \frac{e(1/\lambda) - \beta}{e(1/\lambda) - 1}$$

Similarly in the case of a big update we have

$$\Delta P = \frac{e(\lambda) - \beta}{e(\lambda) - 1}$$

and $\Delta D = 1$ which gives a ratio of

$$\frac{\Delta P}{\Delta D} = \frac{e(\lambda) - \beta}{e(\lambda) - 1}$$

We can conclude by Lemma 19 (inequality (7)) that the ratio of primal cost increase vs dual cost increase is always bounded by

$$\frac{\Delta P}{\Delta D} \leqslant \frac{e(\lambda) - \beta}{e(\lambda) - 1}$$

Using Lemma 10 along with weak duality is enough to conclude that the cost of the fractional solution built by the algorithm is bounded as follows

$$cost_{\mathcal{PDLA}}(\mathcal{A}, \mathcal{I}, \lambda) \leqslant \frac{(e(\lambda) - \beta) \cdot (1 + (1 - \beta)/B)}{e(\lambda) - 1} \cdot \text{OPT}$$

which ends the proof. $\qquad\square$

For consistency, we analyze the algorithm's cost in two parts. When the heuristic algorithm $\mathcal{A}$ buys its $i$th Bahncard at some time $t_i$, define the interval $I_i = [t_i, t_i + T]$ which represents the set of times during which this specific Bahncard is valid. This creates a family of intervals $I_1, \ldots I_k$ if $\mathcal{A}$ buys $k$ Bahncards. Note that we can assume that all these intervals are disjoint since if the prediction $\mathcal{A}$ suggests to buy a new Bahncard before the previous one expires, it is always better to postpone this buy to the end of the validity of the current Bahncard.

**Lemma 12.** *Denote by $(\Delta P)_{I_i}$ the increase in the primal cost of Algorithm 8 during interval $I_i$ and by $cost(\mathcal{A})_{I_i}$ what prediction $\mathcal{A}$ pays during this same interval $I_i$ (including the buy of the Bahncard at the beginning of the interval $I_i$). Then, for all $i$ we have*

$$\frac{(\Delta P)_{I_i}}{cost(\mathcal{A})_{I_i}} \leqslant \frac{\left\lceil \lambda \cdot \frac{B}{1-\beta} \right\rceil}{B + \beta \cdot \left\lceil \lambda \cdot \frac{B}{1-\beta} \right\rceil} \cdot \frac{e(\lambda) - \beta}{e(\lambda) - 1}$$

*Proof.* Assume that $m$ trips are requested during this interval $I_i$. Then we first have that $cost(\mathcal{A})_{I_i} = B + \beta m$ ($\mathcal{A}$ buys a Bahncard then pays a discounted price for every trip in the interval $I_i$).

As for Algorithm 8, for each trip $j$, we are possibly in the first two cases: either $\sum_{t=t(j)-T}^{t(j)} x_t \geqslant 1$ in which case the increase in the primal is $\Delta P = \beta$ or in the second case in which case the increase in the primal is

$$\Delta P = (1-\beta) \cdot \left( \sum_{t=t(j)-T}^{t(j)} x_t + \frac{1}{e(\lambda)-1} \right) + \beta \cdot \sum_{t=t(j)-T}^{t(j)} x_t + 1 - \sum_{t=t(j)-T}^{t(j)} x_t = 1 + \frac{1-\beta}{e(\lambda)-1}$$

We claim that the updates of the second case can happen at most $\left\lceil \lambda \cdot \frac{B}{1-\beta} \right\rceil$ times during interval $I_i$. To see this, denote by $S(l)$ the value of $\sum_{t' \geqslant t_i} x_{t'}$ after $l$ big updates in interval $I_i$. Note that once $\sum_{t' \geqslant t_i} x_{t'} \geqslant 1$, big updates cannot happen anymore. Hence all we need to prove is that $S\left( \left\lceil \lambda \cdot \frac{B}{1-\beta} \right\rceil \right) \geqslant 1$.

We prove by induction that

$$S(k) \geqslant \frac{e(k \cdot (1-\beta)/B) - 1}{e(\lambda)-1}$$

This is indeed true for $k = 0$ as $S(0)$ is the value of $\sum_{t' \geqslant t_i} x_{t'}$ before any big update was made in $I_i$ hence $S(0) \geqslant 0$. Now assume this is the case for some $k$ and compute

$$
\begin{aligned}
S(k+1) &\geqslant \left( 1 + \frac{1-\beta}{B} \right) \cdot S(k) + \frac{1-\beta}{B} \cdot \frac{1}{e(\lambda)-1} \\
&\geqslant \frac{\left( 1 + \frac{1-\beta}{B} \right) \cdot (e(k \cdot (1-\beta)/B) - 1) + \frac{1-\beta}{B}}{e(\lambda)-1} \\
&\geqslant \frac{e((k+1) \cdot (1-\beta)/B) - 1}{e(\lambda)-1}
\end{aligned}
$$

which concludes the induction.

Hence on interval $I_i$, the total increase in the cost of the solution can be bounded as follows

$$(\Delta P)_{I_i} \leqslant \min \left\{ \left\lceil \lambda \cdot \frac{B}{1-\beta} \right\rceil, m \right\} \cdot \left( 1 + \frac{1-\beta}{e(\lambda)-1} \right) + \max \left\{ 0, \left( m - \left\lceil \lambda \cdot \frac{B}{1-\beta} \right\rceil \right) \right\} \cdot \beta$$

One can see that the worst case possible for the ratio $\frac{(\Delta P)_{I_i}}{\text{cost}(\mathcal{A})_{I_i}}$ is obtained for $m = \left\lceil \lambda \cdot \frac{B}{1-\beta} \right\rceil$ and is bounded by

$$\frac{(\Delta P)_{I_i}}{\text{cost}(\mathcal{A})_{I_i}} \leqslant \frac{\left\lceil \lambda \cdot \frac{B}{1-\beta} \right\rceil \cdot \left( 1 + \frac{1-\beta}{e(\lambda)-1} \right)}{B + \beta \cdot \left\lceil \lambda \cdot \frac{B}{1-\beta} \right\rceil} = \frac{\left\lceil \lambda \cdot \frac{B}{1-\beta} \right\rceil}{B + \beta \cdot \left\lceil \lambda \cdot \frac{B}{1-\beta} \right\rceil} \cdot \frac{e(\lambda)-\beta}{e(\lambda)-1}$$

$\qquad\qquad\qquad\qquad\qquad\qquad\qquad\qquad\qquad\qquad\qquad\qquad\qquad\qquad\qquad\qquad\qquad\qquad\quad$ $\square$

We then consider times $t$ that do not belong to any interval $I_i$. More precisely, we upper bound the value $(\Delta P)_j$ that is the increase in cost of the primal solution due to trip $j$ such that $t(j)$ does not belong to any interval $I_i$. Note that in this case the prediction always pays a cost of 1.

**Lemma 13.** *For any trip $j$ such that $t(j) \notin \bigcup_i I_i$, we have that*

$$(\Delta P)_j \leqslant \frac{e(1/\lambda) - \beta}{e(1/\lambda) - 1}$$

*Proof.* Note that Algorithm 8 pays either the cost of a small update which is $\frac{e(1/\lambda)-\beta}{e(1/\lambda)-1}$ or the cost of a minimal update which is $\beta$. $\qquad\qquad\qquad\qquad\qquad\qquad\qquad\qquad\qquad\qquad\qquad$ $\square$

For simplicity and better readability, we will formulate the final theorem of this section only for $\frac{B}{1-\beta} \to \infty$.

**Theorem** (Theorem 4 restated). *For any $\lambda \in (0,1]$, any $\beta \in [0,1]$ and $\frac{B}{1-\beta} \to \infty$, we have the following guarantees on any instance $\mathcal{I}$*

$$cost_{\mathcal{PDLA}}(\mathcal{A}, \mathcal{I}, \lambda) \leqslant \min\left\{\frac{\lambda}{1-\beta+\lambda\beta} \cdot \frac{e^\lambda - \beta}{e^\lambda - 1} \cdot S(\mathcal{A}, I), \frac{e^\lambda - \beta}{e^\lambda - 1} \cdot \mathrm{OPT}\right\}$$

*Proof.* By taking the limit in Lemma 11, we see that the cost of the solution output by Algorithm 8 is at most $\frac{e^\lambda - \beta}{e^\lambda - 1} \cdot \mathrm{OPT}$ which proves the second bound in the theorem.

For the first bound, note that we can write the final cost of the solution as

$$cost_{\mathcal{PDLA}}(\mathcal{A}, \mathcal{I}, \lambda) = \Delta P = \sum_i (\Delta P)_{I_i} + \sum_{j:t(j)\notin \bigcup_i I_i} (\Delta P)_j$$

By taking the limit in Lemma 12 we get that

$$\sum_i (\Delta P)_{I_i} \leqslant \frac{\lambda}{1-\beta+\beta\lambda} \cdot \frac{e^\lambda - \beta}{e^\lambda - 1} \cdot \sum_i cost(\mathcal{A})_{I_i}$$

and by taking the limit in Lemma 13, we get that

$$\sum_{j:t(j)\notin \bigcup_i I_i} (\Delta P)_j \leqslant \frac{e^{1/\lambda} - \beta}{e^{1/\lambda} - 1} \cdot \sum_{j:t(j)\notin \bigcup_i I_i} cost(\mathcal{A})_j$$

By using Lemma 19 (inequality (6)), we see that

$$\max\left\{\frac{\lambda}{1-\beta+\beta\lambda} \cdot \frac{e^\lambda - \beta}{e^\lambda - 1}, \frac{e^{1/\lambda} - \beta}{e^{1/\lambda} - 1}\right\} = \frac{\lambda}{1-\beta+\beta\lambda} \cdot \frac{e^\lambda - \beta}{e^\lambda - 1}$$

which ends the proof.

$\square$

We finish this section by proving that a fractional solution can be rounded online into a randomized integral solution. The expected cost of the rounded instance will be equal to the cost of the fractional solution. If the rounding is very similar to the existing rounding of Buchbinder et al. [5] for ski rental or TCP acknowledgement, we still include it here for completeness as the Bahncard problem was never solved in a primal-dual way. The argument is summarized in the following lemma.

**Lemma 14.** *Given a fractional solution $(x, d, f)$ to the Bahncard problem, it can be rounded online into an integral solution of expected cost equal to the fractional cost of $(x, d, f)$.*

*Proof.* Choose some real number $p$ uniformly at random in the interval $[0,1]$. Then arrange the variables $x_t$ on the real line (i.e. iteratively as follows, each time $t$ takes an interval $I_t$ of length $x_t$ right after the interval taken by $x_{t-1}$). Then buy a Bahncard at every time $t$ such that the interval corresponding to time $t$ contains the real number $p + k$ for some integer $k$. We check first that the expected buying cost is

$$B \cdot \sum_t \mathbb{E}\left(\mathbb{1}_{p+k \in I_t}\right) = B \cdot \sum_t x_t$$

Next, to compute the total expected price of the tickets, notice that if a ticket was bought in the previous $T$ time steps, we can pay a discounted price, otherwise we need to pay the full price of 1. For a trip $j$, the probability that a ticket was bought in the previous $T$ time steps is at least $\sum_{t=t(j)-T}^{t(j)} x_t$. Hence with probability at least $\sum_{t=t(j)-T}^{t(j)} x_t \geqslant d_j$ we pay a price of $\beta$ and with probability $1 - d_j \leqslant f_j$ we pay a price of 1 which ends the proof. $\square$

# C Missing proofs for TCP

## C.1 Plots of instances

We briefly show in Figures 6, 7, and 8 how typical instances under various distributions look like.

Figure 6: Typical instance under Poisson distribution

Figure 7: Typical instance under Pareto distribution

Figure 8: Typical instance under iterated Poisson distribution

## C.2 Theoretical analysis

Recall that we define in this section $e(z) = (1 + 1/d)^{z \cdot d}$ which will be roughly equal to $e^z$ for big $d$. The big updates are then the updates where $z$ is set to $\lambda$ and during a small update, $z$ is set to $1/\lambda$.

We first analyze the consistency of this algorithm. To this end denote by $n_{\mathcal{A}}$ the number of acknowledgements sent by $\mathcal{A}$ and by *latency* $(\mathcal{A})$ the latency paid by the prediction $\mathcal{A}$.

**Lemma 15.** *For any $\lambda \in (0, 1]$, $d > 0$,*

$$c_{\mathcal{PDLA}}(\mathcal{A}, \mathcal{I}, \lambda) \leqslant n_{\mathcal{A}} \cdot \frac{1}{d} \cdot \frac{\lceil \lambda d \rceil}{1 - e(-\lambda)} + latency(\mathcal{A}) \cdot \frac{1}{1 - e(-1/\lambda)}$$

*Proof.* We will use a charging argument to analyze the performance of Algorithm 6. Note that for a small update, the increase in cost of the fractional solution is

$$\Delta P = \frac{1}{d} \left( 1 - \sum_{k=t(j)}^{t} x_k \right) + \frac{1}{d} \cdot \left( \sum_{k=t(j)}^{t} x_k + \frac{1}{e(1/\lambda) - 1} \right) = \frac{1}{d} \cdot \frac{1}{1 - e(-1/\lambda)}$$

However, for every small update that is made, it must be that $\mathcal{A}$ pays a latency of at least $\frac{1}{d}$. Hence the total cost of small updates made by Algorithm 6 is at most $latency(\mathcal{A}) \cdot \frac{1}{1 - e(-1/\lambda)}$.

Secondly we bound the total cost of big updates of our algorithm. Let $t_0$ be a time at which $\mathcal{A}$ sends an acknowledgment. Let $Y$ be the set of big updates made because of jobs $j$ that are acknowledged at time $t_0$ by $\mathcal{A}$ (these big updates are hence made at some time $t \geqslant t_0$). We claim that $|Y| \leqslant \lceil \lambda d \rceil$.

To prove this denote by $S(l)$ the value of $\sum_{k=t_0}^{+\infty} x_k$ after $l$ such big updates (there might be small updates influencing this value but only to make it bigger). Notice that once $\sum_{k=t_0}^{+\infty} x_k \geqslant 1$ there is no remaining update in $Y$. We prove by induction that

$$S(l) \geqslant \frac{(1 + 1/d)^l - 1}{(1 + 1/d)^{\lambda d} - 1}$$

This is clear for $l = 0$ as $S(0) \geqslant 0$. Now assume this is the case for some value $l$ and apply a big update at time $t$ for job $j$ to get

$$
\begin{aligned}
S(l+1) &= S(l) + \frac{1}{d} \cdot \left( \sum_{k=t(j)}^{t} x_k + \frac{1}{e(\lambda) - 1} \right) \\
&\geqslant S(l) \cdot (1 + 1/d) + \frac{1}{d(e(\lambda) - 1)} \\
&= \frac{(1 + 1/d)^{l+1} - 1 - 1/d}{(1 + 1/d)^{\lambda d} - 1} + \frac{1/d}{(e(\lambda) - 1)} \\
&= \frac{(1 + 1/d)^{l+1} - 1 - 1/d}{(1 + 1/d)^{\lambda d} - 1} + \frac{1/d}{(1 + 1/d)^{\lambda d} - 1} \\
&= \frac{(1 + 1/d)^{l+1} - 1}{(1 + 1/d)^{\lambda d} - 1}
\end{aligned}
$$

Where the second inequality comes from noting that since we are considering an update due to a request $j$ acknowledged at time $t_0$ by the predicted solution, it must be that $t(j) \leqslant t_0$ and $\sum_{k=t(j)}^{t} x_k \geqslant \sum_{k=t_0}^{t} x_k$. Hence we get that $S(\lceil \lambda d \rceil) \geqslant 1$ which implies that $|Y| \leqslant \lceil \lambda d \rceil$.

By a similar calculation as for the small update case, we have that the cost of a big update is

$$\Delta P = \frac{1}{d} \cdot \frac{1}{1 - e(-\lambda)}$$

Hence the total cost of these updates in $Y$ is charged to the acknowledgement that $\mathcal{A}$ pays at time $t_0$ to finish the proof. □

Taking the limit $d \to +\infty$ we get the following corollary:

**Corollary 16.** *For any $\lambda \in (0, 1]$ and taking $d \to +\infty$, we have that*

$$c_{\mathcal{PDLA}}(\mathcal{A}, \mathcal{I}, \lambda) \leqslant n_{\mathcal{A}} \cdot \frac{\lambda}{1 - e^{-\lambda}} + latency(\mathcal{A}) \cdot \frac{1}{1 - e^{-1/\lambda}}$$

We then prove robustness of the algorithm with the following lemmas.

**Lemma 17.** *Let $y$ be the dual solution produced by Algorithm 6. Then $\frac{y}{1+1/d}$ is feasible.*

*Proof.* Notice that the constraints of the second type (i.e. $0 \leqslant y_{jt} \leqslant 1/d$) are always satisfied since $0 < \lambda \leqslant 1$. We now check that the second constraints are almost satisfied (within some factor $(1 + 1/d)$). Fix a time $t \in T$ and consider the corresponding constraint:

$$\sum_{j | t \geqslant t(j)} \sum_{t' \geqslant t} y_{jt} \leqslant 1$$

Note that for a small update for some job $j$ such that $t(j) \leqslant t$ the sum above increases by $\lambda/d$ while it increases by $1/d$ for a big update. Notice that once we have that $\sum_{t' \geqslant t} x_{t'} \geqslant 1$, no more such update will be performed. Denote by $S$ the value of this sum.

Notice that for a big update, the sum $S$ becomes $\left(1 + \frac{1}{d}\right) \cdot S + \frac{1}{d((1+1/d)^{\lambda d} - 1)}$. Similarly, for a small updates it becomes $\left(1 + \frac{1}{d}\right) \cdot S + \frac{1}{d((1+1/d)^{d/\lambda} - 1)}$.

Hence, if we denote by $s$ the number of small updates in this sum and by $b$ the number of big updates, by Lemma 20 we have that if $\lambda s + b \geqslant d$ then $\sum_{t' \geqslant t} x_{t'} \geqslant 1$. This directly implies that the value of $\sum_{j | t \geqslant t(j)} \sum_{t' \geqslant t} y_{jt}$ is at most $1 + 1/d$ at the end of the algorithm (each update in the dual is of value at most $1/d$).

Therefore scaling down all $y_{jt}$ by a multiplicative factor of $1 + 1/d$ yields a feasible solution to the dual. □

**Lemma 18.** *For $d \to +\infty$, Algorithm 6 outputs a solution of cost at most $\frac{1}{1-e^{-\lambda}} \cdot \text{OPT}$*

*Proof.* We first compare the increase $\Delta P$ in the primal value to the increase $\Delta D$ in the dual value at every update. We claim that for every update we have

$$\frac{\Delta P}{\Delta D} \leqslant \frac{1}{1 - e(-\lambda)}$$

In the case of a big update we directly have $\Delta P = \frac{1}{d}\left(1 + \frac{1}{e(\lambda) - 1}\right) = \frac{1}{d} \cdot \frac{1}{1 - e(-\lambda)}$ and $\Delta D = \frac{1}{d}$.

In the case of a small update we have $\Delta D = \frac{\lambda}{d}$ and $\Delta P = \frac{1}{d}\left(1 + \frac{1}{e(1/\lambda) - 1}\right) = \frac{1}{d} \cdot \frac{1}{1 - e(-1/\lambda)}$ and we conclude applying Lemma 19 (inequality (3)) that we always have

$$\frac{\Delta P}{\Delta D} \leqslant \frac{1}{1 - e(-\lambda)}$$

By lemma 17, $\frac{y}{1+1/d}$ is a feasible solution. Hence taking $d \to +\infty$ together with the previous remark and weak duality we get the result. □

Combining Lemma 16 and Lemma 18 yields Theorem 5.

# D Optimality bound

**Lemma 3.** *Any $\frac{\lambda}{1-e^{-\lambda}}$-consistent learning augmented algorithm for ski rental has robustness $R(\lambda) \geqslant \frac{1}{1-e^{-\lambda}}$*

*Proof.* For simplicity, we will consider the ski-rental problem in the continuous case which corresponds to the behaviour of the discrete version when $B \to \infty$. In this problem, the cost of buying is 1 and a randomized algorithm has to define a (buying) probability distribution $\{p_t\}_{t \geqslant 0}$. Moreover, consider the case where the true number of vacation days $t_{end} \in [0, 1] \cup (2, \infty)$. In such a case we can assume w.l.o.g. that $p_t = 0, \forall t > 1$. Indeed moving buying probability mass from any $p_t, t > 1$ to $p_1$ does not increase the cost of the randomized algorithm. Assume now that the prediction suggests us that the end of vacations is at $\hat{t}_{end} > 2$, thus the optimal offline solution, if the prediction is correct, is to buy the skis in the beginning for a total cost of 1. Since the algorithm has to define a probability distribution in $[0, 1]$, $\{p_t\}$ needs to satisfy the equality constraint $\int_0^1 p_t dt = 1$. Moreover, note that when the prediction is correct, i.e. $t_{end} > 2$, the LA algorithm suffers an expected cost of $\int_0^1 (t + 1)p_t dt$ while the optimum offline has a cost of 1. Thus the consistency requirement forces the distribution to satisfy the inequality $\int_0^1 (t + 1)p_t dt \leqslant \frac{\lambda}{1-e^{-\lambda}}$. Now assume that the best possible LA algorithm is $c$-robust. If $t_{end} \leqslant 1$ then the LA algorithm's cost is $\int_0^{t_{end}} (t + 1)p_t dt + t_{end} \int_{t_{end}}^1 p_t dt$ while the optimum offline cost is $t_{end}$. Thus, due to $c$-robustness we have that for every $t' \in [0, 1]$, $\int_0^{t'} (t + 1)p_t dt + t' \int_{t'}^1 p_t dt \leqslant ct'$. We calculate the best possible robustness $c$ with the following LP:

Figure 9: Primal Robustness for ski-rental problem.

| **Primal** |
|---|
| minimize $c$ |
| subject to: $\int_0^1 p_t dt = 1$ |
| $\int_0^1 (t + 1)p_t dt \leqslant \frac{\lambda}{1-e^{-\lambda}}$ |
| $\int_0^{t'} (t + 1)p_t dt + t' \int_{t'}^1 p_t dt \leqslant ct' \quad \forall t' \in [0, 1]$ |
| $p_t \geqslant 0 \quad \forall t' \in [0, 1]$ |

To lower bound the best possible robustness $c$ we will present a feasible solution to the dual of 9. The dual variables $\lambda_d$ and $\lambda_c$ correspond respectively to the first and second primal constraints in Figure 9. The dual variables $\lambda_t, \forall t \in [0, 1]$ correspond to the robustness constraints described in the third line of the primal.

The corresponding dual is:

Figure 10: Dual Robustness for ski-rental problem.

| **Dual** |
|---|
| maximize $\lambda_d - \lambda_c \cdot \frac{\lambda}{1-e^{-\lambda}}$ |
| subject to: $\int_0^1 t\lambda_t dt \leqslant 1$ |
| $\lambda_d - (t' + 1)\lambda_c \leqslant \int_0^{t'} t\lambda_t dt + (t' + 1)\int_{t'}^1 \lambda_t dt \quad \forall t' \in [0, 1]$ |
| $\lambda_c, \lambda_t \geqslant 0 \quad \forall t \in [0, 1]$ |

Let $K = \frac{1}{1-\lambda e^{-\lambda}-e^{-\lambda}}$. Then, $\lambda_t = K \cdot e^{-t} \cdot \mathbb{1}\{t \leqslant \lambda\}$, $\lambda_d = K$ and $\lambda_c = K \cdot e^{-\lambda}$.

We first prove that this dual solution is feasible. For the first constraint notice that

$$\int_0^1 t\lambda_t dt = K \cdot \int_0^\lambda te^{-t}dt = K \cdot \left(1 - (\lambda + 1)e^{-\lambda}\right) = 1$$

For the second type of constraint first in the case $t' > \lambda$ we get

$$\int_0^{t'} t\lambda_t dt + (t' + 1)\int_{t'}^1 \lambda_t dt = \int_0^\lambda t\lambda_t dt = 1$$

and we note that

$$\lambda_d - (t'+1)\lambda_c \leqslant \lambda_d - (\lambda+1)\lambda_c = K \cdot \left(1 - (\lambda+1)e^{-\lambda}\right) = 1$$

hence these constraints are satisfied.

In the second case $t' \leqslant \lambda$, we have that

$$
\begin{aligned}
\int_0^{t'} t\lambda_t dt + (t'+1)\int_{t'}^1 \lambda_t dt &= K \cdot \left(\int_0^{t'} te^{-t}dt + (t'+1)\int_{t'}^\lambda e^{-t}dt\right) \\
&= K \cdot \left(1 - (t'+1)e^{-t'} + (t'+1)(e^{-t'} - e^{-\lambda})\right) \\
&= K \cdot \left(1 - (t'+1)e^{-\lambda}\right) \\
&= \lambda_d - (t'+1)\lambda_c
\end{aligned}
$$

which proves that these constraints are also satisfied. Hence this dual solution is feasible. Finally note that the cost of this dual solution is

$$
\begin{aligned}
\lambda_d - \lambda_c \cdot \frac{\lambda}{1 - e^{-\lambda}} &= K \cdot \left(1 - \frac{\lambda}{1 - e^{-\lambda}} \cdot e^{-\lambda}\right) \\
&= K \cdot \frac{1 - e^{-\lambda} - \lambda e^{-\lambda}}{1 - e^{-\lambda}} = \frac{1}{1 - e^{-\lambda}}
\end{aligned}
$$

By weak duality, we conclude that the best robustness cannot be better than $\frac{1}{1-e^{-\lambda}}$ □

# E   Technical lemmas

A few inequalities that will be useful:

**Lemma 19.** *For any $d > 0$, any $0 < \lambda \leqslant 1$, and any $\beta \in [0,1]$, we have:*

$$\frac{\lambda}{1 - e^{-\lambda}} \geqslant \frac{1}{1 - e^{-1/\lambda}} \tag{2}$$

$$\frac{\lambda}{1 - (1 + 1/d)^{-\lambda d}} \geqslant \frac{1}{1 - (1 + 1/d)^{-d/\lambda}} \tag{3}$$

$$\frac{1}{e^\lambda - 1} \geqslant \frac{\frac{1-\lambda}{\lambda} \cdot e^{1/\lambda} + 1}{e^{1/\lambda} - 1} \tag{4}$$

$$\frac{1}{(1 + 1/d)^{\lambda d} - 1} \geqslant \frac{\frac{1-\lambda}{\lambda} \cdot (1 + 1/d)^{d/\lambda} + 1}{(1 + 1/d)^{d/\lambda} - 1} \tag{5}$$

$$\frac{\lambda}{1 - \beta + \beta\lambda} \cdot \frac{e^\lambda - \beta}{e^\lambda - 1} \geqslant \frac{e^{1/\lambda} - \beta}{e^{1/\lambda} - 1} \tag{6}$$

$$(\lambda + \beta - \beta\lambda) \cdot \frac{e^\lambda - \beta}{e^\lambda - 1} \geqslant \frac{e^{1/\lambda} - \beta}{e^{1/\lambda} - 1} \tag{7}$$

*Proof.* Since the formal proof of (2) and (4) seems to require heavy calculations and that they are easy to check on computer we will only give a proof by a plot (see Figures 11a and 11b). For 11b, note that (4) $\iff \frac{1}{e^\lambda - 1} - \frac{1-\lambda}{\lambda} - \frac{1}{\lambda} \cdot \frac{e^{-1/\lambda}}{1 - e^{-1/\lambda}} \geqslant 0$.

(a) Plot of $\frac{\lambda}{1-e^{-\lambda}} - \frac{1}{1-e^{-1/\lambda}}$

(b) Plot of $\frac{1}{e^{\lambda}-1} - \frac{1-\lambda}{\lambda} - \frac{1}{\lambda} \cdot \frac{e^{-1/\lambda}}{1-e^{-1/\lambda}}$

Figure 11: Plots for (2) and (4)

We now prove that inequality (2) implies inequality (3). For this end notice that we can write $(1 + 1/d)^d = e^x$ for some $x \in (0, 1)$ since $(1 + 1/d)^d \in (1, e)$ for all $d > 0$. We prove that for any $x \in (0, 1]$

$$\frac{\lambda\left(1 - e^{-x/\lambda}\right)}{1 - e^{-x\lambda}} \geqslant \frac{\lambda\left(1 - e^{-1/\lambda}\right)}{1 - e^{-\lambda}}$$

which will imply our claim since by inequality (2) the right hand side is bigger than 1. First note this is equivalent to prove that

$$g_\lambda(x) = (1 - e^{-\lambda}) \cdot (1 - e^{-x/\lambda}) - (1 - e^{-1/\lambda}) \cdot (1 - e^{-x\lambda}) \geqslant 0$$

Taking the derivative of $g_\lambda(x)$ we obtain

$$g'_\lambda(x) = \frac{1 - e^{-\lambda}}{\lambda} \cdot e^{-x/\lambda} - \lambda(1 - e^{-1/\lambda}) \cdot e^{-x\lambda}$$

hence we can write

$$g'_\lambda(x) \geqslant 0 \iff e^{x(\lambda - 1/\lambda)} \geqslant \lambda^2 \cdot \frac{1 - e^{-1/\lambda}}{1 - e^{-\lambda}}$$

Notice that the left hand side in this inequality is decreasing because $\lambda \in (0, 1]$. Also notice that $g_\lambda(0) = g_\lambda(1) = 0$. These two facts together imply that $g_\lambda$ is first increasing for $x \in (0, c]$ then decreasing for $x \in (c, 1]$ for some unknown $c$. In particular, we indeed have that $g_\lambda(x) \geqslant 0$ which ends the proof of inequality (3).

Similarly, we prove that inequality (4) implies inequality (5). Again we write $(1 + 1/d)^d = e^x$ for some $x \in (0, 1)$. We first rewrite inequality (5).

$$(5) \iff \frac{1}{e^{\lambda x} - 1} \geqslant \frac{\frac{1-\lambda}{\lambda} \cdot e^{x/\lambda} + 1}{e^{x/\lambda} - 1}$$

$$\iff \frac{1}{e^{\lambda x} - 1} \geqslant \frac{\frac{1-\lambda}{\lambda} \cdot (e^{x/\lambda} - 1) + \frac{1}{\lambda}}{e^{x/\lambda} - 1}$$

$$\iff \lambda(e^{x/\lambda} - 1) \geqslant (1 - \lambda)(e^{x/\lambda} - 1)(e^{\lambda x} - 1) + (e^{\lambda x} - 1)$$

$$\iff \lambda(e^{x/\lambda} - 1) - (1 - \lambda)(e^{x/\lambda} - 1)(e^{\lambda x} - 1) - (e^{\lambda x} - 1) \geqslant 0$$

Define the following function $h_\lambda(x) = \lambda(e^{x/\lambda} - 1) - (1 - \lambda)(e^{x/\lambda} - 1)(e^{\lambda x} - 1) - (e^{\lambda x} - 1)$. One can first compute:

$$h'_\lambda(x) = e^{x/\lambda} - (1-\lambda) \cdot \left( \lambda e^{\lambda x}(e^{x/\lambda} - 1) + \frac{1}{\lambda} e^{x/\lambda}(e^{\lambda x} - 1) \right) - \lambda e^{\lambda x}$$

$$= e^{x/\lambda} - \lambda e^{\lambda x} - (1-\lambda) \cdot \left( (\lambda + 1/\lambda) e^{x(\lambda + 1/\lambda)} - \lambda e^{\lambda x} - \frac{1}{\lambda} e^{x/\lambda} \right)$$

$$= e^{x/\lambda} \cdot \left( 1 + \frac{1-\lambda}{\lambda} \right) + e^{\lambda x} \cdot (-\lambda + \lambda(1-\lambda)) - e^{x(\lambda + 1/\lambda)} \cdot (1-\lambda) \cdot \left( \lambda + \frac{1}{\lambda} \right)$$

$$= \frac{e^{x/\lambda}}{\lambda} - \lambda^2 e^{\lambda x} - \frac{e^{x(\lambda + 1/\lambda)}}{\lambda} \cdot (1-\lambda) \cdot (\lambda^2 + 1)$$

Hence we can rewrite

$$h'_\lambda(x) \geqslant 0 \iff \frac{e^{x/\lambda}}{\lambda} - \lambda^2 e^{\lambda x} - \frac{e^{x(\lambda + 1/\lambda)}}{\lambda} \cdot (1-\lambda) \cdot (\lambda^2 + 1) \geqslant 0$$

$$\iff e^{x/\lambda} - \lambda^3 e^{\lambda x} - e^{x(\lambda + 1/\lambda)} \cdot (1-\lambda) \cdot (\lambda^2 + 1) \geqslant 0$$

$$\iff 1 - \lambda^3 e^{x(\lambda - 1/\lambda)} - e^{x\lambda} \cdot (1-\lambda) \cdot (\lambda^2 + 1) \geqslant 0$$

Let us define $i_\lambda(x) = 1 - \lambda^3 e^{x(\lambda - 1/\lambda)} - e^{x\lambda} \cdot (1-\lambda) \cdot (\lambda^2 + 1)$ and we derive

$$i'_\lambda(x) = -\lambda^3 \cdot (\lambda - 1/\lambda) \cdot e^{x(\lambda - 1/\lambda)} - \lambda e^{\lambda x} \cdot (1-\lambda) \cdot (\lambda^2 + 1)$$

We can now notice that

$$i'_\lambda(x) \geqslant 0 \iff -\lambda^3 \cdot (\lambda - 1/\lambda) \cdot e^{x(\lambda - 1/\lambda)} - \lambda e^{\lambda x} \cdot (1-\lambda) \cdot (\lambda^2 + 1) \geqslant 0$$

$$\iff -\lambda^3 \cdot (\lambda - 1/\lambda) \cdot e^{-x/\lambda} - \lambda(1-\lambda) \cdot (\lambda^2 + 1) \geqslant 0$$

$$\iff \lambda^3 \cdot (1/\lambda - \lambda) \cdot e^{-x/\lambda} - \lambda(1-\lambda) \cdot (\lambda^2 + 1) \geqslant 0$$

Since the left hand side is decreasing as $x$ increases we only need to check one extreme value which is $i'_\lambda(0)$. We write

$$i'_\lambda(0) \leqslant 0 \iff \lambda^3 \cdot (1/\lambda - \lambda) - \lambda \cdot (1-\lambda) \cdot (\lambda^2 + 1) \leqslant 0$$

$$\iff \lambda^2 - \lambda^4 - (\lambda^3 + \lambda - \lambda^4 - \lambda^2) \leqslant 0$$

$$\iff -\lambda^3 + 2\lambda^2 - \lambda \leqslant 0$$

$$\iff -\lambda \cdot (\lambda - 1)^2 \leqslant 0$$

hence we always have $i'_\lambda(0) \leqslant 0$.

Therefore we get that $i'_\lambda(x) \leqslant 0$ for all $x$ and $\lambda$. Note that $i_\lambda(0) = 1 - \lambda^3 - (1-\lambda)(\lambda^2 + 1) = 1 - \lambda^3 - \lambda^2 - 1 + \lambda^3 + \lambda = \lambda - \lambda^2 \geqslant 0$. Therefore we get that $h_\lambda$ is first positive on some interval $[0, c]$ and then negative for $x \in [c, \infty)$. Therefore $h_\lambda$ is first increasing then decreasing. Notice that $h_\lambda(0) = 0$ and $h_\lambda(1) \geqslant 0$ by inequality (4). Hence inequality (5) is true for all $x \in [0, 1]$ which concludes the proof.

Finally, the proof of (6) and (7) are quicker and similar. Note that

$$(6) \iff \lambda \cdot \frac{e^\lambda - \beta}{e^\lambda - 1} \geqslant (1 - \beta + \beta\lambda) \cdot \frac{e^{1/\lambda} - \beta}{e^{1/\lambda} - 1}$$

which is equivalent to a polynomial (in $\beta$) of degree 2 being positive. The leading coefficient of this polynomial $P$ is negative and we notice that $P(1) = 0$ and that $P(0) \geqslant 0$ by (2). All these facts together imply that $P(\beta) \geqslant 0$ for all $\beta \in [0, 1]$. The proof of (7) is similar. $\qquad\square$

**Lemma 20.** *Let $0 < \lambda \leqslant 1$, $d > 0$ and define the following functions ($x \in \mathbb{R}$):*

$$f(x) = \left( 1 + \frac{1}{d} \right) \cdot x + \frac{1}{d \left( (1 + 1/d)^{\lambda d} - 1 \right)}$$

$$g(x) = \left(1 + \frac{1}{d}\right) \cdot x + \frac{1}{d\left((1 + 1/d)^{d/\lambda} - 1\right)}$$

*Given $S \geqslant 0$ and a word $w \in \{a, b\}^*$ we define a sequence $S_w$ recursively as follows:*

$$S_{w.y} = \begin{cases} S \text{ if } w.y = \varepsilon \\ f(S_w) \text{ if } y = a \\ g(S_w) \text{ if } y = b \end{cases}$$

*Then for any $w \in \{a, b\}^*$ such that $|w|_a + \lambda |w|_b \geqslant d$ we have that $S_w \geqslant 1$.*

*Proof.* Let $w' = b \ldots b a \ldots a = b^{|w|_b} a^{|w|_a}$ be the word made of $|w|_b$ consecutive $b$s followed by $|w|_a$ consecutive $a$s. Then we claim that $S_{w'} \leqslant S_w$. This directly follows from the fact that for any real number $x$, $f(g(x)) \leqslant g(f(x))$. Noticing this, we can swap positions between an $a$ followed by a $b$ and reducing the final value. We keep doing this until all the $b$s in $w$ end up in front position.

With standard computations one can check that

$$S_{b^{|w|_b}} = S \cdot (1 + 1/d)^{|w|_b} + \frac{(1 + 1/d)^{|w|_b} - 1}{(1 + 1/d)^{d/\lambda} - 1}$$

For ease of notation define $S' = S_{b^{|w|_b}}$. Using the assumption that $|w|_a + \lambda |w|_b \geqslant d$ and that $S \geqslant 0$ we get that

$$S' \geqslant \frac{(1 + 1/d)^{(d - |w|_a)/\lambda} - 1}{(1 + 1/d)^{d/\lambda} - 1}$$

Again using standard calculations we get that

$$S_{w'} \geqslant S' \cdot (1 + 1/d)^{|w|_a} + \frac{(1 + 1/d)^{|w|_a} - 1}{(1 + 1/d)^{\lambda d} - 1}$$

which implies

$$S_{w'} \geqslant \frac{(1 + 1/d)^{(d - |w|_a)/\lambda} - 1}{(1 + 1/d)^{d/\lambda} - 1} \cdot (1 + 1/d)^{|w|_a} + \frac{(1 + 1/d)^{|w|_a} - 1}{(1 + 1/d)^{\lambda d} - 1}$$

Define $h(x) = \frac{(1+1/d)^{(d-x)/\lambda} - 1}{(1+1/d)^{d/\lambda} - 1} \cdot (1 + 1/d)^x + \frac{(1+1/d)^x - 1}{(1+1/d)^{\lambda d} - 1}$. We finish the proof by proving that for any $0 < \lambda \leqslant 1$, any $d > 0$ and any $x \geqslant 0$, we have that $h(x) \geqslant 1$.

Note that $h(0) = 1$ and that

$$h'(x) = \ln(1 + 1/d) \cdot \left( \frac{(1 + 1/d)^x}{(1 + 1/d)^{\lambda d} - 1} - \frac{1 - \lambda}{\lambda} \cdot \frac{(1 + 1/d)^{(d - (1-\lambda)x)/\lambda}}{(1 + 1/d)^{d/\lambda} - 1} - \frac{(1 + 1/d)^x}{(1 + 1/d)^{d/\lambda} - 1} \right)$$

To study the sign of $h'(x)$ we can drop the $\ln(1 + 1/d)$ and write

$$h'(x) \geqslant 0 \iff \frac{(1 + 1/d)^x}{(1 + 1/d)^{\lambda d} - 1} - \frac{1 - \lambda}{\lambda} \cdot \frac{(1 + 1/d)^{(d - (1-\lambda)x)/\lambda}}{(1 + 1/d)^{d/\lambda} - 1} - \frac{(1 + 1/d)^x}{(1 + 1/d)^{d/\lambda} - 1} \geqslant 0$$

$$\iff \frac{1}{(1 + 1/d)^{\lambda d} - 1} - \frac{1 - \lambda}{\lambda} \cdot \frac{(1 + 1/d)^{(d-x)/\lambda}}{(1 + 1/d)^{d/\lambda} - 1} - \frac{1}{(1 + 1/d)^{d/\lambda} - 1} \geqslant 0$$

Clearly the last term is increasing as $x$ increases hence we can limit ourselves to prove that $h'(0) \geqslant 0$ which we can rewrite

$$h'(0) \geqslant 0 \iff \frac{1}{(1+1/d)^{\lambda d} - 1} - \frac{1-\lambda}{\lambda} \cdot \frac{(1+1/d)^{d/\lambda}}{(1+1/d)^{d/\lambda} - 1} - \frac{1}{(1+1/d)^{d/\lambda} - 1} \geqslant 0$$

$$\iff \frac{1}{(1+1/d)^{\lambda d} - 1} \geqslant \frac{\frac{1-\lambda}{\lambda} \cdot (1+1/d)^{d/\lambda} + 1}{(1+1/d)^{d/\lambda} - 1}$$

Which holds by equation (5) of Lemma 19. $\qquad\qquad\qquad$ □