[Reviews · NeurIPS 2020]

Review 1

Summary and Contributions: The paper proposes and analyzes primal-dual algorithms for a number of online optimization problems, which are augmented by an arbitrary solution prediction. Given a parameter lambda, which controls the trust in the prediction, the authors provide performance bounds on the obtained solutions based on both the optimal offline solution and the predicted solution. The theoretical results are complemented by experiments for one of the problems, which illustrate the behavior of the method.

Strengths: The theoretical results appear to be novel for all presented problems and are enough material for a journal publication on the topic. From what I understand (not being an expert), this work significantly extends the results on online optimization, by incorporating predictions in the primal-dual method. The results seem very valuable to online optimization community.

Weaknesses: I do not see major weaknesses in this work. --- Update --- After the rebuttal, my initial assessment remains.

Correctness: I could not check all technical details (e.g. Lemma 20) due to limited time. I checked the proofs of the main claims, however, and I could not find major flaws. Minor comments are included below.

Clarity: The paper is clearly structured and generally well written. The accessibility of the proofs could be improved a bit at times by defining important quantities outside of the text.

Relation to Prior Work: The related work is discussed along with each presented problem. The differences are laid out clearly. Another section discussing the commonalities of the presented problems together with the related work may be desirable. I am not an expert for this type of problems, however.

Reproducibility: Yes

Additional Feedback: The proof sketches might be valuable for readers very familiar when the presented techniques. For others it may be better to just discuss the main ideas underlying all presented results and then make sure that the main proofs are easy to read. Please properly define the value d in line 111. The informal definition used in the text is unclear and misleading. Questions about proofs: Please clarify where you assume that w_S >= 1 (line 449). Why does Alg 4 stop after at most N updates if N^{pred} <= B (proof of Thm 2) ? In the proof of Lem 15 between line 651 and 652, the second equality is actually an inequality. Also clarify why the prior inequality holds. Typos: "banhcard" line 201 summation index lines 647-648


Review 2

Summary and Contributions: The authors study an adaptation of the primal-dual method that incorporates predictions. The show a (somewhat) black box way of converting algorithms that were designed using the primal dual method using a covering LP to include predictions, which are in the form of the actual prediction A and a 'confidence' \lambda in [0,1]. Using this, they augment previously known state-of-the-art online algorithms for Set Cover, Ski Rental, Bahncard Problem and Dynamic TCP acknowledgement to use advice and are able to upper bound their competitive ratio by two terms: one which includes only \lambda and the ex-post error if the advice is followed blindly and one which includes \lambda and the optimal algorithm, therefore achieving a trade-off that between using the advice, but never straying too far from the worst case, even if the advice is useless. To briefly summarize the primal-dual method for covering problems: a covering problem is one whose offline version can be solved using a covering LP, i.e. one where all constraints are of the form A*X >= B where A, B contains positive coefficients. To design the online algorithm, the LP is initially almost empty but at every turn, a new set of constraints is added while the variables stay the same. Then, some variables are continuously increased until the new constraint is met, while also keeping the increase of the dual variables as small as possible to obtain a guarantee by weak duality. The type of constraints is very important: having only non-negative coefficients can capture problems where different sets of resources can be spent at different rates (set cover, ski rental etc) but usually not ones where the solution can move 'back-and-forth', such as the k-server problem. In their approach the actual LP is not modified, however when updating the variables each time a constraint is added the change can be biased to move slightly faster towards the advice. This change usually doesn't disrupt the primal-dual proof, so a slightly weaker bound against the optimal algorithm is easy, while at the same time a guarantee against following the advice blindly can be obtained using a slightly ad-hoc, charging argument. Finally, the authors have included experiments that showcase that the performance is indeed better for small inputs, and is not drowned by hidden factors or only holds asymptotically.

Strengths: 1) The paper provides a generalized way of adapting a specific class of online algorithms to incorporate advice. 2) The actual settings studied in the paper are interesting in their own right and are not just toy problems. 3) The paper managed to fit enough interesting content in 8 pages, even though it's theoretical. Someone could decide if this approach works for them without having to go through the full version.

Weaknesses: 1) I am not sure I would call the approach black box in the strict sense, as there is still a fair amount of tailor made analysis for each setting (and the algorithm modification, although intuitive, still requires a deep understanding of the problem). 2) Covering problems are generally very well understood in the online literature. Are there any interesting covering problems left for others to apply this technique?

Correctness: I did not read everything in the supplementary material, but the main lemmata and theorems statements follow what would be expected for an online paper using the primal dual method.

Clarity: Yes, it was fun to read.

Relation to Prior Work: Yes.

Reproducibility: Yes

Additional Feedback: After reading the rebuttal and discussing with the other reviewers, I am confident that this paper is a very nice result indeed.


Review 3

Summary and Contributions: This paper presents a new version of the Primal-Dual algorithm for solving online set-covering problems, which is able to take into account a solution predicted by any other algorithm. The Primal-Dual algorithm for online problems builds fractional solutions by augmenting the primal and the dual solutions each time a new item to be covered becomes available (the problem is considered online). The proposed modification for handling predicted solution consists basically of augmenting the variable of the primal solutions using a convex combination of the increment given by the original Primal-Dual algorithm (see Algorithm 2), and a quantity which consider the predicted integral solution given in input to the new algorithm. The constant (parameter) \lambda of the convex combination, which is indeed in the interval [0,1], should reflect how much we trust the predicted solution: for \lambda=0 the online algorithm will exactly build the predicted solution; for \lambda=1, the algorithm will produce the same solution of the original Primal-Dual algorithm. The proposed algorithm is detailed for three different problems, which are all a variant of the online set covering problem: the ski rental problem, the Banchard problem, and a Dynamic TCP acknowledgment problem. For each variant of the set covering problem, the authors provide a formal bound obtained with their algorithm, which is a specialization of their Theorem 1.

Strengths: The main advantage of this paper is that it presents a simple and intuitive idea for building fractional solutions for online problems which can benefit from a predicted solution. In addition, the authors prove a formal bound for each variant of the set covering problem considered. The results given in the theorems are intuitive and the proofs look correct.

Weaknesses: The main weakness of this paper is that the three problems considered are very similar, and the computational results are limited and somehow artificial. Moreover, it inherits the same issues of the Primal-Dual algorithm: it only build fractional solutions, hence, we are still left with the problem of building online an integral solution. Little comments are provided with respect with this issue. The computational results are limited and they are restricted only to the last problem (Dynamic TCP Acknowledgement).

Correctness: The results (the five theorems in the paper) looks correct, and they generalize previous results on standard online algorithms.

Clarity: All the algorithms could be better presented, for instance, by reporting only Algorithms 2, 4, and 6 with the line numbers, and then by specifying which lines are present only in the Augmented version (and by omitting the description for Algorithms 1, 3, and 5). This way the authors could save space to better explain how to build integral solutions. Otherwise, the paper is well written, but a general conclusion is missing.

Relation to Prior Work: The paper clearly presents related works and it discusses how it generalize previous results, appeared recently in the ML literature, as for instance [25].

Reproducibility: Yes

Additional Feedback: UPDATE AFTER REBUTTAL: We are thankful to the authors for their precise answer. After reading the rebuttal and the comments of the other reviewers, we have increased the overall score. --------------------------------------------------------------------------------------------------- The main questions for the authors are: - how do you recover optimal integral solutions, from the fractional solutions you get with your algorithms? - do standard dataset exist for benchmarking online algorithms, likely with data coming from real industrial applications? - could the authors benefit from distribution errors coming from the predictors?

[Author Response · NeurIPS 2020]

We would like to thank the reviewers for their detailed reading of our work and constructive comments. We will take into account all their valuable remarks on the general presentation and readability of the paper. We now address their more specific concerns separately for each reviewer in order of appearance.

**Reviewer #1.** We will clarify the definition of $d$ in line 111 by defining $d = \max_{e \in \mathcal{U}} |\mathcal{F}(e)|$ in a separate equation.

*Please clarify where you assume that $w_S >= 1$ (line 449).* We use this assumption in the inequality just below where we can assume that $(1 + 1/w_S)^{w_S} \geq 2$ since $w_S \geq 1$.

*Why does Alg 4 stop after at most N updates if $N^{pred} <= B$ (proof of Thm 2) ?* Alg 4 always stops after at most N updates, regardless of the prediction. This is because the true instance contains N skiing days so N new constraints are revealed online. After the last constraint has arrived, the algorithm always stops.

*In the proof of Lem 15 between line 651 and 652, the second equality is actually an inequality. Also clarify why the prior inequality holds.* Agreed, the second equality is an inequality. For the prior inequality, recall that $S(k)$ is the value of $\sum_{l=t_0}^{\infty} x_l$ at time $t$ after $k$ big updates. Thus, $S(k) = \sum_{l=t_0}^{t} x_l$. Since we are considering an update due to a request $j$ acknowledged at time $t_0$ by the predicted solution, it must be that $t(j) \leq t_0$ hence we have that $\sum_{l=t(j)}^{t} x_l \geq \sum_{l=t_0}^{t} x_l$. Noting this, we can lower bound the first line replacing $\sum_{l=t(j)}^{t} x_l$ by $\sum_{l=t_0}^{t} x_l = S(k)$.

**Reviewer #2.** *I am not sure I would call the approach black box in the strict sense, as there is still a fair amount of tailor made analysis for each setting (and the algorithm modification, although intuitive, still requires a deep understanding of the problem).* The term black-box was used to underline the fact that the designed algorithms do not make any assumption on the quality of the prediction and use it in a "black-box manner". We will clarify this by avoiding the term black-box in the final version of the paper. Moreover, we agree that modifying the primal dual algorithm requires a deep understanding of the algorithm itself. However, for all problems we considered, only the additive term in the update of the primal variables needs to be modified (in a similar vein). Hence it is true that the approach is not a black box in the strict sense but the basic idea of the framework is a great help in obtaining the result.

*Covering problems are generally very well understood in the online literature. Are there any interesting covering problems left for others to apply this technique?* There are actually a lot of covering problems that are solved via the online primal-dual technique like for instance the weighted caching and load balancing problems (see [4] in our bibliography). In addition to that, we suspect that our work might also provide insights not only for covering problems but also for packing problems. A concrete example of that is the revenue maximization in ad auctions problem which also has an optimal $\frac{e}{e-1}$-competitive algorithm and whose dual is similar to the TCP acknowledgement problem (see [5]). Moreover, another interesting direction may be to incorporate predictions into the primal dual technique when it is used to solve covering problems where the objective function is non linear (e.g. convex). We will add a separate paragraph discussing these future directions in the final version of the paper.

**Reviewer #3.** *how do you recover optimal integral solutions, from the fractional solutions you get with your algorithms?* We did not emphasize this point since it already appeared in previous literature (except for the Bahncard problem). However, we agree that this is a very crucial point and deserves more emphasis. To obtain our results, we only modify the algorithm that builds the fractional solution but not the rounding scheme. For the ski rental, Bahncard and TCP acknowledgement problems it is possible to round (randomly) the solution in an online manner such that the expected cost of the integral solution obtained is equal to the cost of the fractional solution. The rounding scheme for ski rental and TCP can be found in the reference [5] and in Lemma 14 of our paper for the Bahncard problem. For set cover a multiplicative $O(\log n)$ factor is lost during the rounding due to the integrality gap of the linear program. The rounding scheme for the online set cover problem is described by Alon et al. in [1].

*do standard datasets exist for benchmarking online algorithms, likely with data coming from real industrial applications?* To the best of our knowledge there are no standard datasets to benchmark online algorithms. In the new field of learning augmented online algorithms the only datasets used so far were Brighkite and CitiBikeNYC by Lykouris and Vassilvitskii in [19]. The aforementioned datasets were not used in our work because the access requests were too sparse to model the TCP requests of a large server. Although we could not find a real dataset we would like to underline that our artificial dataset was created to mimic a real TCP packet instance following common assumptions regarding packet distribution as noted in [11,20,22,30].

*could the authors benefit from distribution errors coming from the predictors?* We would like to emphasize that the main challenge that we address is to design a learning augmented algorithm which uses the prediction without any assumption on its quality and on the type of prediction errors. That being said, once additional assumptions are made, it is very possible that one can twist our results to obtain better guarantees. We believe that this is an intriguing direction with possibly interesting results. However, this direction is complementary to the area of learning augmented algorithms as it approaches the problem under a stochastic optimization point of view.

[Meta-Review · NeurIPS 2020]

The paper investigates a general method to augment primal-dual algorithms for incorporating predictions. The paper presents a nice theoretical and practical mix of results for a number of online optimization problems that opens possibilities to generally augment online algorithms. Reviewers agree on the rich contribution and varied content and generally good exposition. The rebuttal addressed reviewers' concerns adequately. This is a clear accept. We urge the authors to enhance the abstract to clarify what is meant by "incorporate predictions" in this context (e.g. "incorporate arbitrary solution-proposing heuristics"). The broader impact section is inadequate and should better explain how the work will help to advance ML in general.